# Histone deacetylase 3 represses cholesterol efflux during CD4+ T-cell activation

Drew Wilfahrt[1], Rachael L Philips[1†], Jyoti Lama[1], Monika Kizerwetter[1‡], Michael Jeremy Shapiro[1], Shaylene A McCue[1], Madeleine M Kennedy[1], Matthew J Rajcula[1], Hu Zeng[1,2], Virginia Smith Shapiro[1*]

[1]Department of Immunology, Mayo Clinic, Rochester, United States; [2]Division of Rheumatology, Department of Medicine, Mayo Clinic, Rochester, United States

*For correspondence:
shapiro.virginia1@mayo.edu

Present address: [†]Molecular Immunology and Inflammation Branch, National Institute of Arthritis and Musculoskeletal and Skin Diseases, National Institutes of Health, Bethesda, United States; [‡]Department of Biomedical Engineering, Johns Hopkins University, Baltimore, United States

Competing interest: The authors declare that no competing interests exist.

**Abstract** After antigenic activation, quiescent naive CD4+ T cells alter their metabolism to proliferate. This metabolic shift increases production of nucleotides, amino acids, fatty acids, and sterols. Here, we show that histone deacetylase 3 (HDAC3) is critical for activation of murine peripheral CD4+ T cells. HDAC3-deficient CD4+ T cells failed to proliferate and blast after in vitro TCR/CD28 stimulation. Upon T-cell activation, genes involved in cholesterol biosynthesis are upregulated while genes that promote cholesterol efflux are repressed. HDAC3-deficient CD4+ T cells had reduced levels of cellular cholesterol both before and after activation. HDAC3-deficient cells upregulate cholesterol synthesis appropriately after activation, but fail to repress cholesterol efflux; notably, they overexpress cholesterol efflux transporters ABCA1 and ABCG1. Repression of these genes is the primary function for HDAC3 in peripheral CD4+ T cells, as addition of exogenous cholesterol restored proliferative capacity. Collectively, these findings demonstrate HDAC3 is essential during CD4+ T-cell activation to repress cholesterol efflux.

## Editor's evaluation

This paper will be of interest to scientists in the field of T cell biology and immunometabolism. The data analysis is rigorous and the experiments performed are appropriate. The findings of the manuscript will expand upon previous findings of a role for histone deacetylase 3 in thymocyte development and CD8+ T cell function to that of CD4+ T cells.

## Introduction

After activation, CD4+ T cells must pass through a number of metabolic checkpoints in order to proliferate, differentiate, and generate robust immune responses. This metabolic transition shortly after TCR engagement has been defined as quiescence exit, and is characterized by several key cellular events including cell growth, interleukin-2 (IL-2) signaling, increased anabolic metabolism, and reprogramming of mitochondrial metabolism (*Chapman et al., 2019*; *Chapman and Chi, 2018*; *Ron-Harel et al., 2016*; *Tan et al., 2017*; *Wang et al., 2011*; *Yang et al., 2013*). Each of these checkpoints prepares CD4+ T cells for proliferation and effector function by generating the cellular building blocks required for activated T cells such as fatty acids and sterols, nucleotides, amino acids, and other metabolites (*Bensinger et al., 2008*; *Geiger et al., 2016*; *Johnson et al., 2018*; *Kidani et al., 2013*; *Ma et al., 2017*; *Ricciardi et al., 2018*; *Wang et al., 2011*). Given the importance of these molecules during activation, this metabolic reprogramming is under the control of keenly regulated molecular

circuits to ensure resources are used efficiently. To date, our understanding of the transcriptional control of CD4[+] T cells exiting quiescence is incomplete.

Mechanistic target of rapamycin (mTOR), particularly mTOR complex 1 (mTORC1), directs many aspects of metabolic reprogramming after T-cell activation (*Tan et al., 2017*; *Yang et al., 2013*). One role of mTORC1 is to drive lipid synthesis through the expression of sterol regulatory element-binding proteins (SREBPs) (*Kidani et al., 2013*). SREBPs are transcription factors that orchestrate lipid synthesis after T-cell activation (*DeBose-Boyd and Ye, 2018*). In addition to increasing lipid and sterol synthesis, recently activated T cells also halt cholesterol efflux (*Bensinger et al., 2008*). Activated T cells rapidly decrease expression of cholesterol efflux transporters in order to retain recently generated cholesterol (*Michaels et al., 2021*). Both of these steps, increased cholesterol synthesis and decreased cholesterol efflux, are required for successful proliferation and blast formation. Disruptions in cholesterol synthesis in CD8[+] T cells inhibited blasting and proliferation after TCR engagement (*Kidani et al., 2013*), while enforced expression of cholesterol efflux transporter ABCG1 inhibited proliferation (*Bensinger et al., 2008*). Cholesterol metabolism may not be identically regulated in CD4[+] and CD8[+] T cells, as deletion of Acetyl-CoA Acetyltransferase 1 (ACAT1) enhanced proliferation and effector function in CD8[+] T cells but not CD4[+] T cells (*Yang et al., 2016*). Together, these studies point to an important 'cholesterol checkpoint' in which T cells require an optimal amount of cholesterol to exit quiescence.

Although previous work highlights the importance of transcription factors in the regulation of cholesterol homeostasis, less is known about the role of chromatin modifiers in the regulation of cholesterol availability in T cells. Histone deacetylase 3 (HDAC3) is a Class I HDAC that deacetylates lysine residues on histones H3 and H4 in order to repress gene expression. Previously, our group has shown that HDAC3 serves as a targeted regulator of key gene expression during T-cell development. During positive selection in the thymus, HDAC3 is required for downregulation of RORγt (*Philips et al., 2016*). Further, HDAC3 suppression of the purinergic-receptor P2RX7 is critical for survival of double positive thymocytes in the ATP-rich thymic cortex (*Philips et al., 2019b*). Recently, the role of HDAC3 as an inhibitor of the cytotoxicity program of CD8[+] T cells was examined using E8I-Cre, which initiates deletion in CD8 SP thymocytes (*Ellmeier et al., 1997*; *Tay et al., 2020*). Collectively, this work supports the idea that HDAC3 is a highly specific transcriptional regulator in lymphocytes.

Little is known of the role that HDAC3 plays in peripheral CD4[+] T cells. Here, we report that HDAC3-deficient CD4[+] T cells have a loss of differentiated helper T-cell populations in vivo. This loss of differentiated T-cell numbers is due to an inability of HDAC3-deficient CD4[+] T cells to blast and proliferate after activation. HDAC3-deficient CD4[+] T cells upregulate cholesterol synthesis genes normally after activation, but fail to downregulate cholesterol efflux. This results in reduced cellular cholesterol levels in HDAC3-deficient T cells before and after T-cell activation. HDAC3-deficient cells upregulate mRNA expression of genes encoding the cholesterol efflux transporters ABCA1 and ABCG1. Increased mRNA expression is maintained after TCR ligation. Further, deletion of HDAC3 results in hyperacetylation of promoter sites for both *Abca1* and *Abcg1*, consistent with direct gene regulation by HDAC3 deacetylase activity. Importantly, the addition of exogenous cholesterol restores proliferative capacity of HDAC3-deficient CD4[+] T cells, indicating that a decreased cholesterol level is the primary block preventing proliferation and blasting. Thus, HDAC3 is required to maintain cholesterol availability after T-cell activation through the repression of cholesterol efflux.

## Results

### CD8[+] T cells have intrathymic deletion of HDAC3 in dLck-Cre HDAC3 cKO, but CD4[+] T cells initiate deletion in recent thymic emigrants

Previous studies have outlined several important roles for HDAC3 during T-cell development in the thymus (*Hsu et al., 2015*; *Philips et al., 2016*; 2019; *Philips et al., 2019a*; *Stengel et al., 2015*). To interrogate the role of HDAC3 in peripheral T cells, distal-Lck-Cre (dLck-Cre) HDAC3 cKO mice were generated. In this system, Cre recombinase expression is driven by the distal promoter of lymphocyte-specific protein tyrosine kinase (Lck). Previous studies showed this system drives Cre expression after positive selection in the thymus (*Zhang et al., 2005*). Adult dLck-Cre HDAC3 cKO mice had normal numbers of naive and memory CD4[+] T-cell populations in the spleen, but had a significant decrease in CD8[+] T-cell populations (*Figure 1a*). Previous work in which HDAC3 was deleted in the thymus

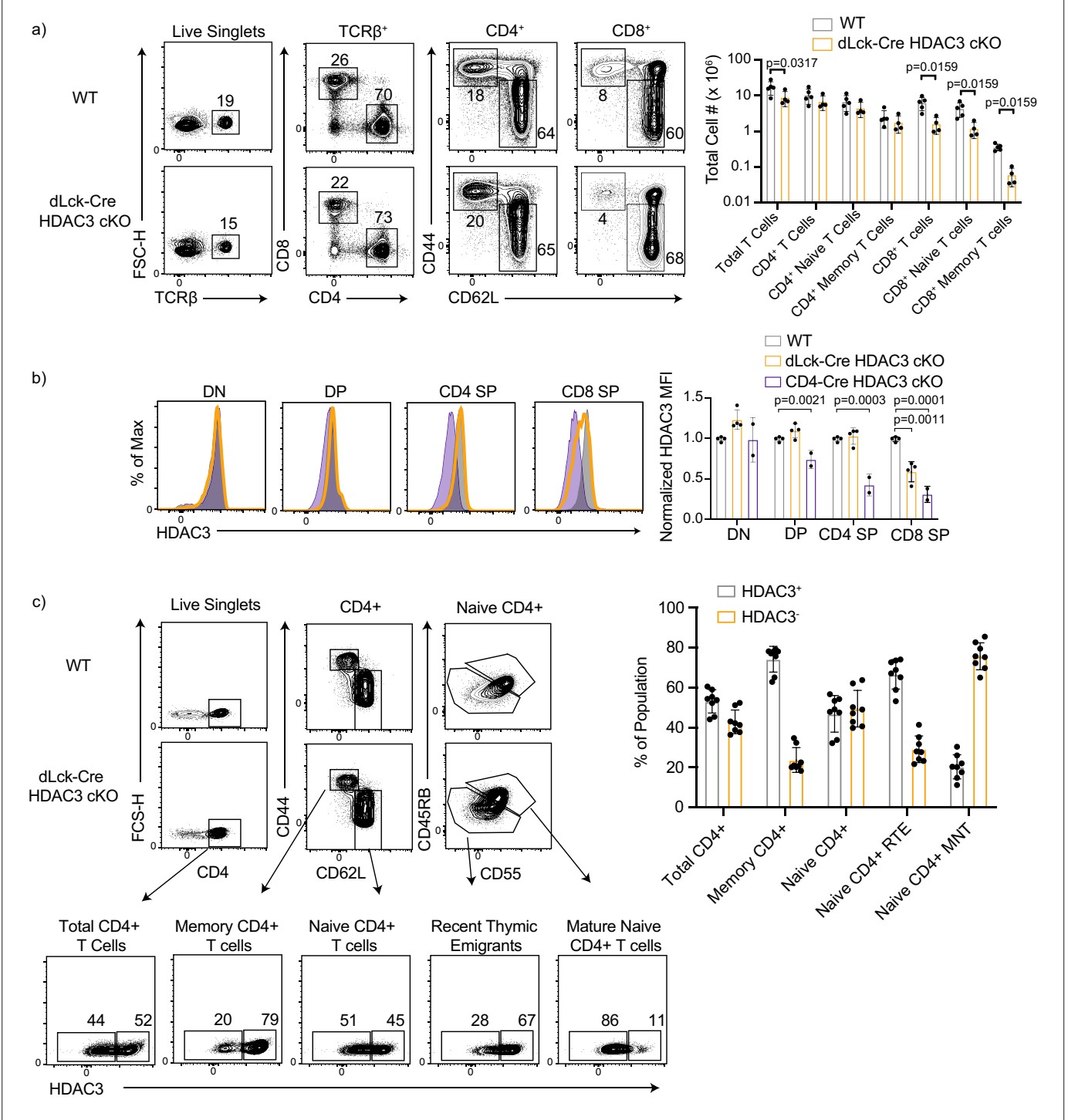

**Figure 1.** CD8[+] T cells have intrathymic deletion of histone deacetylase 3 (HDAC3) in dLck-HDAC3 cKO, but CD4[+] T cells initiate deletion at the recent thymic emigrant (RTE) stage. (**a**) Profile of primary splenic T-cell populations from wild-type (WT) and dLck-Cre HDAC3 cKO mice including total T cells (TCRβ[+]), total CD4[+] (TCRβ[+] CD4[+]), and total CD8[+] (TCRβ[+] CD8[+]), as well as memory (CD44[hi] CD62L[lo]) and naive (CD44[lo] CD62L[hi]) from each of the CD4[+] and CD8[+] populations. Bar graph depicts mean ± standard deviation (SD). Total cell number from three independent experiments (*n* = 5 mice/group). Statistical significance was determined for the indicated comparisons with Mann–Whitney tests between each WT and cKO population. (**b**) Expression of HDAC3 in thymocyte populations from WT, dLck-Cre HDAC3 cKO and CD4-Cre HDAC3 cKO mice. Thymic populations are gated as in *Figure 1— figure supplement 1*, and quantification of normalized HDAC3 MFI (median fluorescent intensity) ± SD from three independent experiments is shown on the right (*n* = 2–4 mice/group). Statistical significance was determined for the indicated comparisons with a one-way analysis of variance (ANOVA) with Tukey's multiple comparisons test. (**c**) Profile of HDAC3 deletion in dLck-Cre HDAC3 cKO mice. Splenocytes were gated on key populations

*Figure 1 continued on next page*

*Figure 1 continued*

including total CD4$^+$ (CD4$^+$), naive CD4$^+$ (CD4$^+$ CD44$^{lo}$ CD62L$^{hi}$), and memory CD4$^+$ (CD4$^+$CD44$^{hi}$CD62L$^{lo}$). Naive cells were further gated in mature naive T cells (MNTs; CD45RB$^{hi}$CD55$^{hi}$) or recent thymic emigrants (RTEs; CD45RB$^{lo}$CD55$^{lo}$). HDAC3-positive and -negative population frequencies are shown below. Bar chart on right quantifies the mean frequency ± SD of HDAC3$^+$ or HDAC3$^-$ events within each population (*n* = 8 mice/group from three independent experiments).

The online version of this article includes the following figure supplement(s) for figure 1:

**Figure supplement 1.** dLck-Cre histone deacetylase 3 (HDAC3) cKO mice have normal thymocyte population numbers.

revealed HDAC3 is required for T-cell maturation, leading to a block at the recent thymic emigrant (RTE) stage (*Hsu et al., 2015*).

Since there were differences in peripheral CD8$^+$ T-cell numbers in the dLck-Cre HDAC3 cKO mice, the kinetics of HDAC3 deletion were investigated to explore the possibility of intrathymic HDAC3 deletion. To do this, developing thymocyte populations were examined. Total numbers of double negative (DN), double positive (DP), and CD4 and CD8 single positive (CD4 SP/CD8 SP) in the dLck-Cre HDAC3 cKO thymus were roughly equivalent to wild-type (WT) mice (*Figure 1—figure supplement 1*). Surprisingly, the CD8 SP population had a loss of HDAC3 protein level compared to WT CD8 SP (*Figure 1b*). Thus, dLck-Cre HDAC3 cKO initiated deletion as early as the CD8 SP thymocyte stage. HDAC3 expression was unaffected in CD4 SP thymocytes. Given the critical roles for HDAC3 in developing thymocytes, we concluded that dLck-Cre HDAC3 cKO mice are not a suitable model for examination of mature peripheral CD8$^+$ T cells in the absence of HDAC3. Thus, this work focuses on the role of HDAC3 in CD4$^+$ T cells.

Having established that HDAC3 expression is intact in the developing CD4$^+$ SP thymocytes, the HDAC3 protein expression in peripheral CD4$^+$ T-cell populations at homeostasis was assessed. In previously published studies with dLck-Cre systems, deletion is inefficient in the mature CD4$^+$ T-cell populations (*Zhang et al., 2005*; *Zhang et al., 2010*). To measure the efficiency of HDAC3 deletion in dLck-Cre HDAC3 cKO mice, naive (CD62L$^{hi}$CD44$^{lo}$) and memory phenotype (CD62L$^{lo}$CD44$^{hi}$) CD4$^+$ T cells were examined. RTEs and mature naive T cells (MNTs) were distinguished using CD55 and CD45RB, both markers that are upregulated during peripheral T-cell maturation. HDAC3 deletion began soon after T cells egress from the thymus since ~25% of RTEs were HDAC3 deficient in the dLck-Cre HDAC3 cKO (*Figure 1c*). Further, MNTs were enriched for HDAC3$^-$ cells, with >75% of them being HDAC3 deficient (*Figure 1c*). Surprisingly, only 25% of the memory CD4$^+$ T cells were HDAC3 deficient (*Figure 1c*) suggesting that HDAC3-deficient memory CD4$^+$ T cells had a competitive disadvantage to the HDAC3-sufficient memory CD4$^+$ cells in dLck-Cre HDAC3 cKO mice.

## HDAC3-deficient CD4$^+$ T cells are capable of differentiation, but produce fewer cells than WT CD4$^+$ T cells

Given the incongruence of HDAC3 deletion between the naive and memory CD4$^+$ T-cell populations in the dLck-Cre HDAC3 cKO, we examined whether HDAC3 could play a role in the formation and expansion of memory CD4$^+$ T cells. To test this, differentiated helper T-cell populations were measured at homeostasis in vivo. There were very few HDAC3-deficient T helper (T$_h$) cells including T$_h$1, T$_h$2, T$_h$17, T$_{reg}$, and T$_{fh}$ cells in the spleen in dLck-Cre HDAC3 cKO mice when compared to WT (*Figure 2*). The frequency of HDAC3-deficient T$_h$17 cells in the mesenteric lymph nodes (mLNs) was also reduced in dLck-Cre HDAC3 cKO mice (*Figure 2—figure supplement 1*). The frequency of the other T$_h$ populations in mLN and Peyer's patches was not statistically different (*Figure 2—figure supplement 1*, *Figure 2—figure supplement 2*). In fact, total numbers of the HDAC3-deficient differentiated splenic populations more closely resembled CD4-Cre HDAC3 cKO mice, which are highly lymphopenic (*Hsu et al., 2015*). Of note, HDAC3-deficient cells could differentiate in a noncompetitive environment. CD4-Cre HDAC3 cKO mice generated T$_h$2, T$_h$17 and T$_{reg}$ and T$_{fh}$ cells at about the same frequency as WT cells in the spleen and the mesenteric lymph node although total numbers were highly reduced (*Figure 2*, *Figure 2—figure supplement 1*). Since those mice are highly lymphopenic, they exist in a relatively noncompetitive environment when compared to the competitive dLck-Cre HDAC3 cKO mice where HDAC3-sufficient cells are present. With this information, we hypothesized that HDAC3 plays a role in maintaining CD4$^+$ T cell fitness to successfully differentiate in vivo.

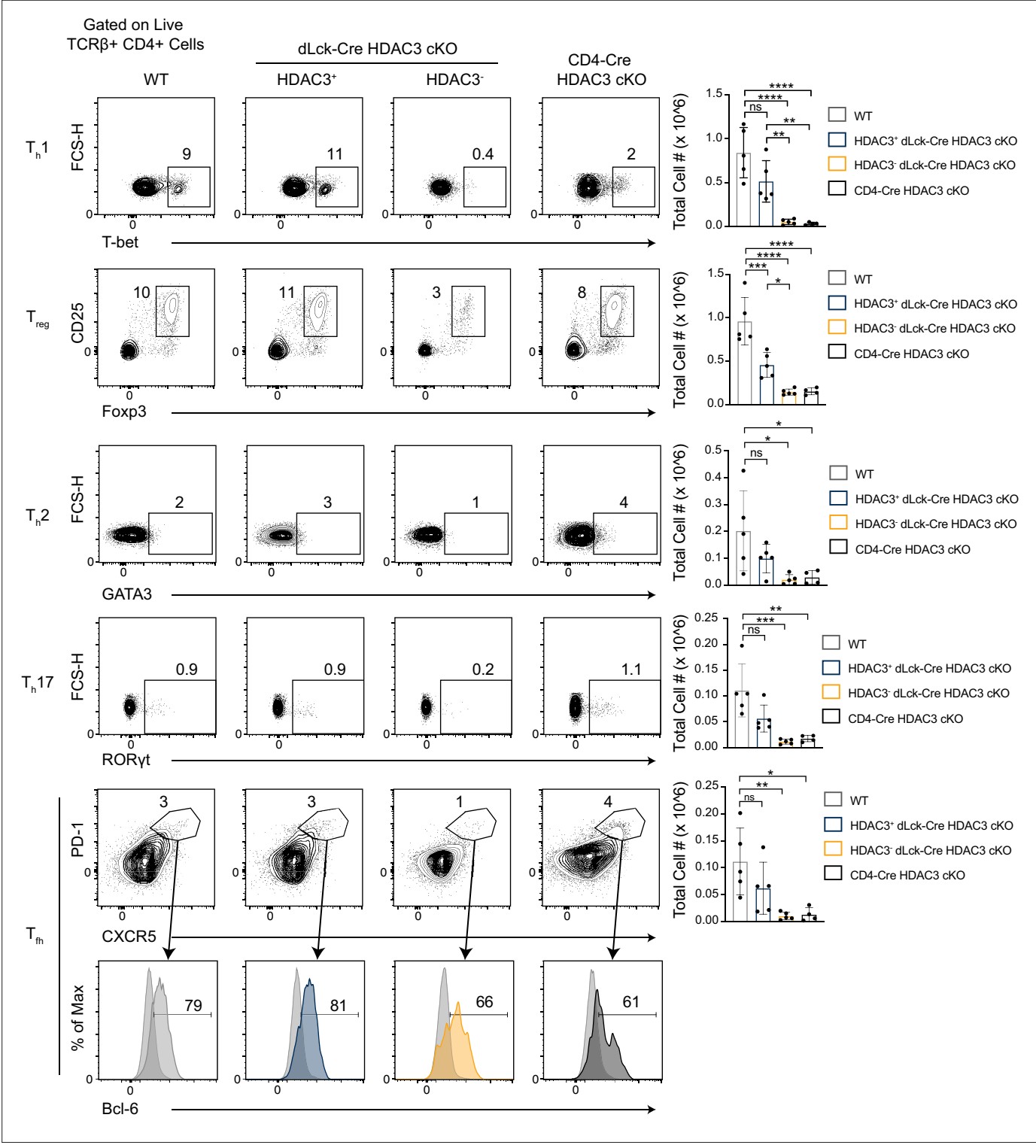

**Figure 2.** Histone deacetylase 3 (HDAC3)-deficient CD4+ T cells from dLck-Cre HDAC3 cKO mice have reduced differentiated $T_h$-cell populations. Identification of helper T-cell populations in vivo. Splenocytes were harvested from wild-type (WT) and HDAC3 cKO mice, and labeled for flow cytometry. Cells were first gated on HDAC3+ or HDAC3− events, then gating for $T_h1$ (T-bet+), $T_h2$ (GATA3+), $T_h17$ (ROR $\gamma$ t+), $T_{reg}$ (Foxp3+ CD25+), and $T_{fh}$ (CXCR5+ PD-1+ Bcl-6hi) is shown (left). Bar plots on right represent pooled data for the total cell number ± standard deviation (SD) from three independent experiments (*n* = 4–5 mice/group in total). Non-$T_{fh}$ CXCR5− PD-1− cells (dark gray histograms) were used as a negative control for Bcl-6 expression to set the gate on the Bcl-6 histograms. Statistical significance was determined for the indicated comparisons using ordinary one-way analysis of variance (ANOVA) with Tukey's multiple comparisons test (*p < 0.05, **p < 0.01, ***p < 0.001, ****p < 0.0001).

*Figure 2 continued on next page*

*Figure 2 continued*

The online version of this article includes the following figure supplement(s) for figure 2:

**Figure supplement 1.** Histone deacetylase 3 (HDAC3)-deficient T cells from dLck-Cre HDAC3 cKO mice have reduced differentiated T$_h$-cell populations.

**Figure supplement 2.** Histone deacetylase 3 (HDAC3)-deficient T cells from Peyer's patches have reduced T$_{fh}$ cell numbers.

To define the importance of HDAC3 in T-cell differentiation, in vitro differentiation assays were performed to differentiate naive CD4$^+$ T cells from WT or dLck-Cre HDAC3 cKO mice to T$_h$1, T$_h$2, T$_h$17, and T$_{reg}$ lineages, and lineage-defining transcription factor and cytokine expression was measured after 4 days. Notably, HDAC3$^-$ T cells from dLck-Cre HDAC3 cKO mice differentiated normally and expressed the transcription factors Foxp3, RORγt, GATA3, and T-bet under appropriate polarizing conditions (*Figure 3*). Likewise, HDAC3$^-$ T cells from dLck-Cre HDAC3 cKO mice had a normal frequency of cells expressing IFN-γ, IL-4, and IL-17A under appropriate conditions and coexpression of each lineage-defining transcription factor and cytokine was observed for each T$_h$ lineage (*Figure 3—figure supplement 1*). Thus, HDAC3 was not required for T$_h$ differentiation in vitro. To further test whether CD4$^+$ T cells from dLck-Cre HDAC3 cKO mice that differentiated in vivo were functionally impaired, magnetically enriched CD4$^+$ T cells from WT and dLck-Cre HDAC3 cKO spleens were stimulated with PMA/ionomycin for 6 hr, and subsequently examined for expression of IFN-γ and T-bet. Consistent with the in vitro differentiation assays, HDAC3-deficient memory CD4$^+$ T cells from dLck-Cre HDAC3 cKO mice had a similar frequency of IFN-γ$^+$ events to the WT, although the frequency of IFN-γ$^+$ T-bet$^{hi}$ cells was reduced among total CD4$^+$ T cells (*Figure 3—figure supplement 2*). Thus, HDAC3-deficient CD4$^+$ T cells that undergo differentiation in vivo are not functionally impaired.

Interestingly, the total number of cells harvested from dLck-Cre HDAC3 cKO mice in the day 4 in vitro differentiation assay cultures was >4-fold lower than WT for each of the T$_h$1, T$_h$2, and T$_h$17 differentiation assays (*Figure 3*, right). The output of Foxp3$^+$ T$_{reg}$ cells in the dLck-Cre HDAC3 cKO culture was not statistically different than WT, and intriguingly, HDAC3$^-$ T cells from the dLck-Cre HDAC3 cKO mice had increased Foxp3 MFI and a higher percentage of cells that were Foxp3$^+$ among all cells in the culture (*Figure 3*). Thus, differentiation as measured by transcription factor expression is not impaired in the HDAC3-deficient T cells, but the cellular output is diminished after T-cell activation.

## HDAC3-deficient CD4$^+$ T cells have reduced proliferation, diminished mTORC1 signaling after in vitro stimulation

Since HDAC3-deficient cells exhibited a reduced cell number in the differentiation assays, proliferation was examined. To test this, CD4$^+$ T cells were magnetically enriched, labeled with CFSE (carboxyfluorescein succinimidyl ester), stimulated with αCD3/αCD28 for 3 days, and examined by flow cytometry. There was a severe defect in the ability of HDAC3-deficient CD4$^+$ T cells to proliferate upon CD3/CD28 stimulation (*Figure 4a*). However, HDAC3-deficient CD4$^+$ T cells did not have a global impairment in their ability to respond to CD3/CD28 stimulation, as induction of CD69 expression was only slightly reduced. Greater than 80% of HDAC3-deficient T cells upregulated CD69 expression, and the MFI was similar between WT and HDAC3-deficient CD4$^+$ T cells, indicating that HDAC3-deficient CD4$^+$ T cells can respond to TCR signals (*Figure 4b*). However, IL-2 expression was reduced by 50 % in HDAC3-deficient T cells compared to WT after αCD3/αCD28 stimulation (*Figure 4c*). In addition, the percentage of HDAC3$^-$ cells that induced CD25 expression 3 days after αCD3/αCD28 stimulation was greatly reduced (*Figure 4d*). Since IL-2R signaling plays a critical role in T-cell proliferation after TCR activation (*Boyman and Sprent, 2012*; *Ross and Cantrell, 2018*), reduced expression of both IL-2 cytokine and IL-2 receptor likely contribute to defective proliferation in the HDAC3$^-$ population. IL-2 also sends key survival signals to activated T cells (*Boyman and Sprent, 2012*; *Ross and Cantrell, 2018*). Cell death was also measured in the HDAC3$^-$ dLck-Cre HDAC3 cKO cells by labeling activated cells with fixable viability dye 24 hr after activation. HDAC3-deficient T cells had a significant decrease in the percentage of viable cells compared to WT (*Figure 4e*). Since, P2RX7 expression was increased in HDAC3-deficient thymocytes (*Philips et al., 2019b*), P2RX7 expression was examined in HDAC3-deficient CD4$^+$ T cells after CD3/CD28 stimulation in vitro. HDAC3-deficient CD4$^+$ T cells did not have a statistically significant change of P2RX7 expression in either naive or memory CD4$^+$ T cells (*Figure 4f*). Altogether, deletion of HDAC3 in peripheral CD4$^+$ T cells results in impaired proliferation

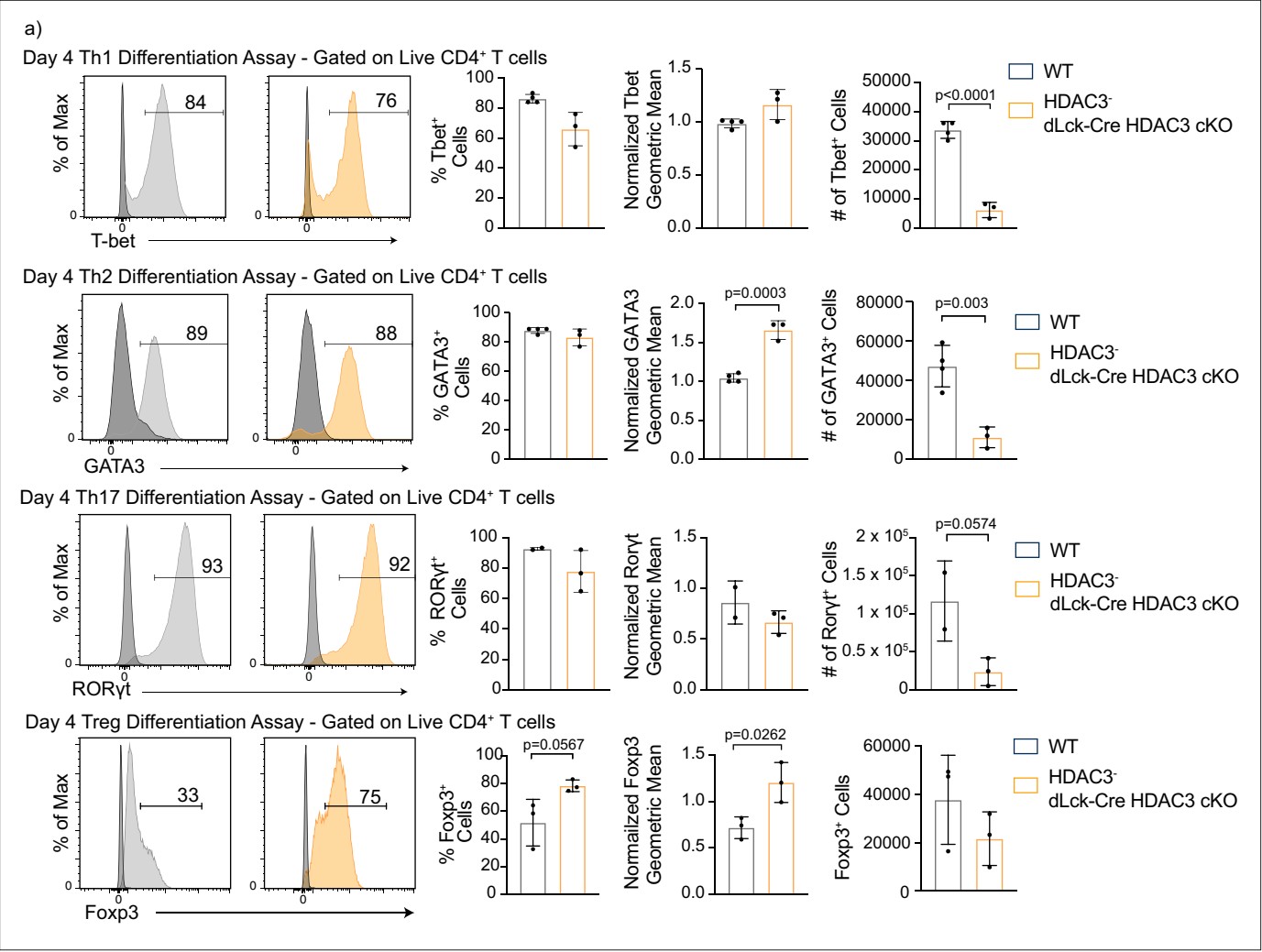

**Figure 3.** Histone deacetylase 3 (HDAC3)-deficient CD4$^+$ T cells are capable of differentiation, but produce fewer cells than wild-type (WT). (**a**) In vitro differentiation assays were performed to examine differentiation into the T$_h$1, T$_h$2, T$_h$17, and T$_{reg}$ lineages characterized by transcription factor expression. Splenocytes were harvested and magnetically enriched for naive (CD44$^-$) CD4$^+$ T cells by negative selection. Cells in all assays were stimulated with 2 µg/ml plate-bound αCD3 and 0.5 µg/ml αCD28 for 4 days. For T$_h$1 differentiation, 1 µg/ml αIL-4 antibody and 10 ng/ml of IL-12 were added to the media. For T$_h$2 differentiation, 1 µg/ml of each αIFN$\gamma$ and αIL-12 antibody, as well as 10 ng/ml of IL-4 was added to the media. For T$_h$17 differentiation, media was supplemented with 10 µg/ml of αIFN$\gamma$ and αIL-4 antibody as well as 10 ng/ml of rIL-23, 5 ng/ml TGF-β1, and 20 ng/ml IL-6. For T$_{reg}$ differentiation, media was supplemented with 10 µg/ml αIFN$\gamma$ and αIL-4 antibody as well as 2 ng/ml TGF-β1, and 2 ng/ml interleukin-2 (IL-2). Unstimulated control samples did not receive αCD3/αCD28 stimulation, but did receive 10 ng/ml IL-7 to maintain cell survival during culture. Bar plots on the right show % of cells from the total culture that are positive for the transcription factor ± standard deviation (SD), total cell number ± SD, and geometric mean of expression ± SD ($n$ = 2–4 mice/group from two to three independent experiments for each). Two nonlittermate, but age and sex matched, WT B6 controls were used in these experiments. Negative controls (black histograms) represent unstimulated, but stained samples from the same mouse as the stimulated sample shown. Statistical significance was determined for the indicated comparisons using an unpaired $t$-test.

The online version of this article includes the following figure supplement(s) for figure 3:

**Figure supplement 1.** Histone deacetylase 3 (HDAC3)-deficient CD4 T cells are capable of producing lineage-specific cytokines during in vitro differentiation.

**Figure supplement 2.** Histone deacetylase 3 (HDAC3)-deficient CD4 T cells are capable of producing IFN$\gamma$ ex vivo.

and survival after activation, suggesting that HDAC3 protein is necessary for successful expansion of CD4$^+$ T-cell populations after TCR engagement.

Since IL-2 signaling, survival and proliferation are key outcomes of successful quiescence exit, HDAC3-deficient CD4$^+$ T cells from dLck-Cre HDAC3 cKO mice were examined for participation in the required metabolic switch during T-cell activation. Since mTORC1 activity orchestrates many of these

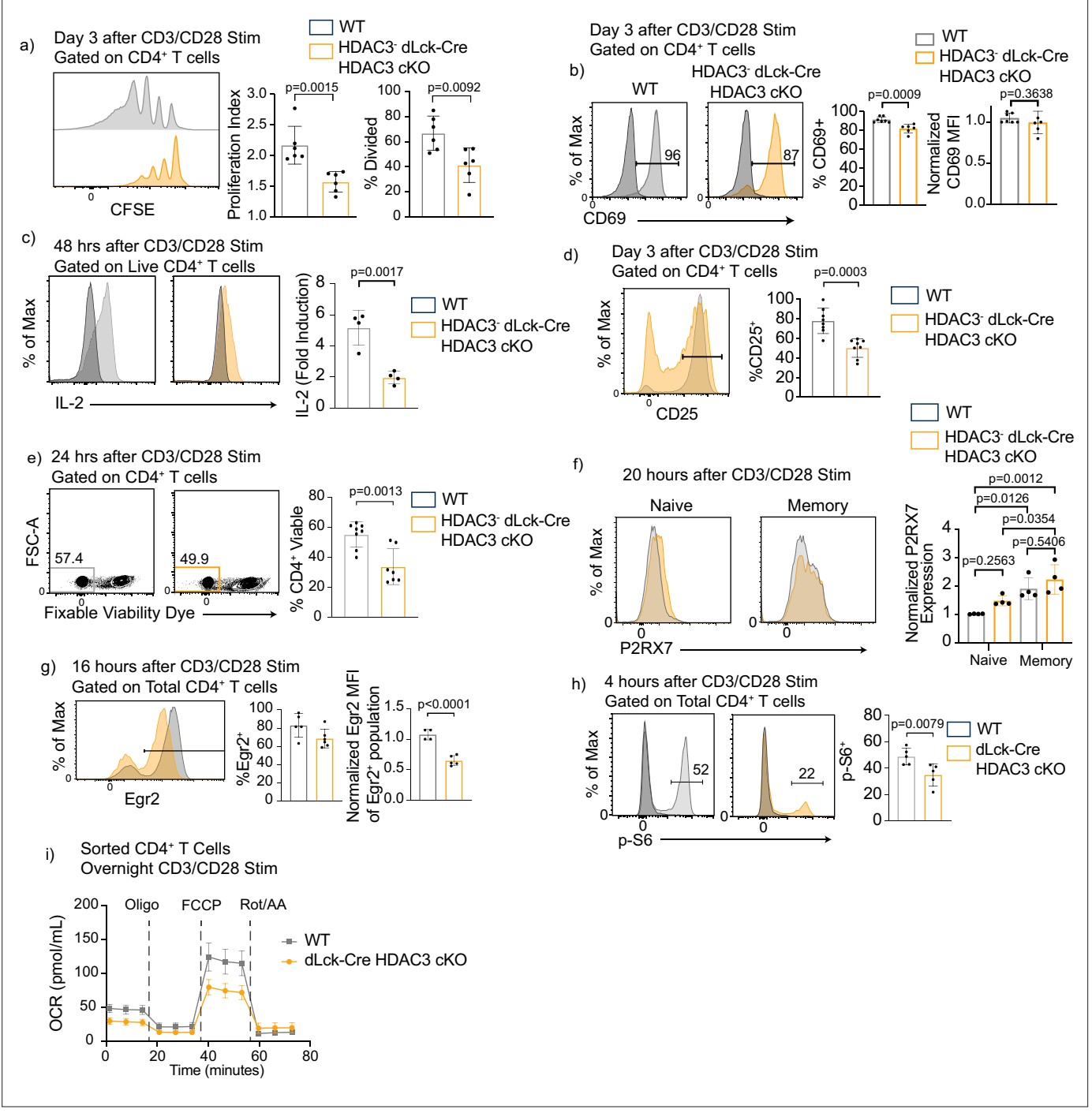

**Figure 4.** Histone deacetylase 3 (HDAC3)-deficient CD4+ T cells have reduced proliferation and diminished mechanistic target of rapamycin (mTOR) signaling after in vitro stimulation. (**a**) Splenocytes were isolated from wild-type (WT) or dLck-Cre HDAC3 cKO mice, and magnetically enriched for CD4+ T cells by magnetic negative selection. Enriched CD4+ T cells were labeled with CFSE and stimulated with 5 μg of plate-bound αCD3 and 1 μg/ ml of αCD28 for 3 days and examined by flow cytometry for proliferation. Cells from dLck-Cre HDAC3 cKO were gated on HDAC3− cells to eliminate contaminating HDAC3+ events. Bar graphs on right show proliferation index ± standard deviation (SD) and % divided ± SD (*n* = 6 mice/group from three independent experiments). One nonlittermate, but age- and sex-matched WT B6 control was used in this experiment. (**b**) Enriched CD4+ T cells were stimulated with 5 μg plate-bound αCD3 and 1 μg/ml soluble αCD28 and analyzed for CD69 expression after 16 hr. Bar graph on right shows MFI ± SD (*n* = 6–7 mice/group from three independent experiments). (**c**) Enriched CD4+ T cells were analyzed for interleukin-2 (IL-2) production 48 hr after stimulation with 2 μg of plate-bound αCD3 and 0.5 μg/ml of αCD28. Bar graph on right shows fold induction ± SD (*n* = 4 mice/group from two independent experiments). (**d**) Enriched CD4+ T cells were stimulated with 5 μg plate-bound αCD3 and 1 μg/ml soluble αCD28 and analyzed for CD25 receptor expression after 3 days. Bar graph on right shows percent CD25 positive ± SD (*n* = 8 mice/group from four independent experiments). (**e**)

*Figure 4 continued on next page*

*Figure 4 continued*

Enriched CD4$^+$ T cells were stimulated with 5 μg plate-bound αCD3 and 1 μg/ml soluble αCD28 and analyzed for viability after 24 hr. Bar graph on right shows percent viability dye positive ± SD ($n$ = 7 mice/group from four independent experiments). (**f**) Enriched CD4$^+$ T cells were stimulated with 5 μg plate-bound αCD3 and 1 μg/ml soluble αCD28 and analyzed for expression of P2RX7 after 20 hr. Bar graph on right shows MFI ± SD ($n$ = 4 mice/group from two independent experiments). (**g**) Enriched CD4$^+$ T cells were stimulated with 5 μg plate-bound αCD3 and 1 μg/ml soluble αCD28 and analyzed for Early Growth Response Protein 2 (Egr2) expression after 16 hr. Bar graphs on right shows percent Egr2$^+$ ± SD and Egr2 MFI of Egr2$^+$ population ± SD ($n$ = 4–5 mice/group from three independent experiments). (**h**) Enriched CD4$^+$ T cells were stimulated with 5 μg plate-bound αCD3 and 1 μg/ml soluble αCD28 and analyzed for phosphorylation of ribosomal protein S6 after 4 hr. Bar graph on right shows percent p-S6 positive ± SD ($n$ = 5 mice/group from three independent experiments). (**i**) Enriched CD4$^+$ T cells were stimulated overnight with 5 μg plate-bound αCD3 and 1 μg/ml soluble αCD28, then a Mito Stress Assay was conducted to measure oxygen consumption rate (OCR). Data are pooled, and show mean OCR ± standard error of the mean (SEM; $n$ = 3 mice/group from two independent experiments). Statistical significance for indicated comparisons in all panels except for (f) was determined by an unpaired $t$-test. For (f), statistical significance was determined for the indicated comparisons using one-way analysis of variance (ANOVA) and with Tukey's multiple comparisons test.

The online version of this article includes the following figure supplement(s) for figure 4:

**Figure supplement 1.** dLck-Cre histone deacetylase 3 (HDAC3) cKO cells have reduced basal extracellular acidification rate after activation.

changes, including lipid synthesis (*Mossmann et al., 2018*), events downstream of mTORC1 activity were investigated. One important mTORC1-S6K1-dependent signaling event is expression of Early Growth Response Protein 2 (Egr2) (*Kurebayashi et al., 2012*). Sixteen hours after αCD3/αCD28 stimulation, CD4$^+$ HDAC3$^-$ T cells from dLck-Cre HDAC3 cKO mice exhibited a comparable percentage of Egr2$^+$ cells with WT CD4$^+$ T cells, but the MFI of HDAC3-deficient cells was significantly reduced (*Figure 4g*). Additionally, dLck-Cre HDAC3 cKO CD4$^+$ T cells had reduced phosphorylation of ribosomal protein S6 (*Figure 4h*), indicating that mTOR activity was diminished in dLck-Cre HDAC3 cKO cells. To more accurately assess the energetic activity of recently activated dLck-Cre HDAC3 cKO T cells, magnetically enriched CD4$^+$ T cells from WT and KO spleens were stimulated with αCD3/αCD28 antibodies overnight, and then a Mito Stress Assay was conducted. The assay revealed the dLck-Cre HDAC3 cKO CD4$^+$ T cells had a decrease in both basal and maximal respiratory capacity compared to WT as measured by oxygen consumption rate (OCR) (*Figure 4i*). Moreover, basal extracellular acidification rate was also reduced in the dLck-Cre HDAC3 cKO cells (*Figure 4—figure supplement 1*). Collectively, these data suggest that dLck-Cre HDAC3 cKO cells have a disruption in the metabolic reprogramming required for successful T-cell activation.

## HDAC3-deficient CD4$^+$ T cells have defective blasting, reduced cholesterol levels, and increased cholesterol efflux transporter expression

mTORC1 activity plays a pivotal role in lipid synthesis after T-cell activation (*Chapman et al., 2019*). As such, we investigated whether dLck-Cre HDAC3 cKO T cells had a phenotype consistent with defective lipid and cholesterol availability. HDAC3-deficient CD4$^+$ T cells from dLck-Cre HDAC3 cKO mice were examined to see whether they were capable of cell growth and blasting after activation, which is dependent upon the production and retention of lipids (*Armstrong et al., 2010*; *Bensinger et al., 2008*; *Kidani et al., 2013*). Three days after αCD3/αCD28 stimulation, HDAC3-deficient CD4$^+$ T cells had a reduced frequency of blasting cells, and their median size measured by forward scatter area (FSC-A) was significantly reduced compared to WT cells (*Figure 5a*). Granularity as measured by side scatter area (SSC-A) was also reduced in the HDAC3-deficient CD4$^+$ T cells (*Figure 5a*). Filipin III, a naturally fluorescent antibiotic that binds to cholesterol and has previously been used as a probe for cellular cholesterol levels (*Muller et al., 1984*), was utilized to determine whether cellular cholesterol levels were altered in HDAC3-deficient T cells. Notably, volume-adjusted Filipin III signal was significantly reduced in HDAC3-deficient CD4$^+$ T cells both before and after a 20-hr stimulation with αCD3/αCD28 (*Figure 5b*), indicating that cellular cholesterol concentrations were decreased in the absence of HDAC3. Since cholesterol is implicated in the formation of lipid rafts, which are key components of T-cell receptor signaling (*Bietz et al., 2017*; *Fessler, 2016*; *Fessler and Parks, 2011*), we utilized a fluorescently conjugated Cholera Toxin Subunit B (CTB) molecule to probe for lipid rafts (*Lencer and Tsai, 2003*). After 20 hr of αCD3/αCD28 stimulation, HDAC3-deficient T cells had a small but significant reduction in CTB fluorescence, consistent with a reduction of lipid raft concentration (*Figure 5c*).

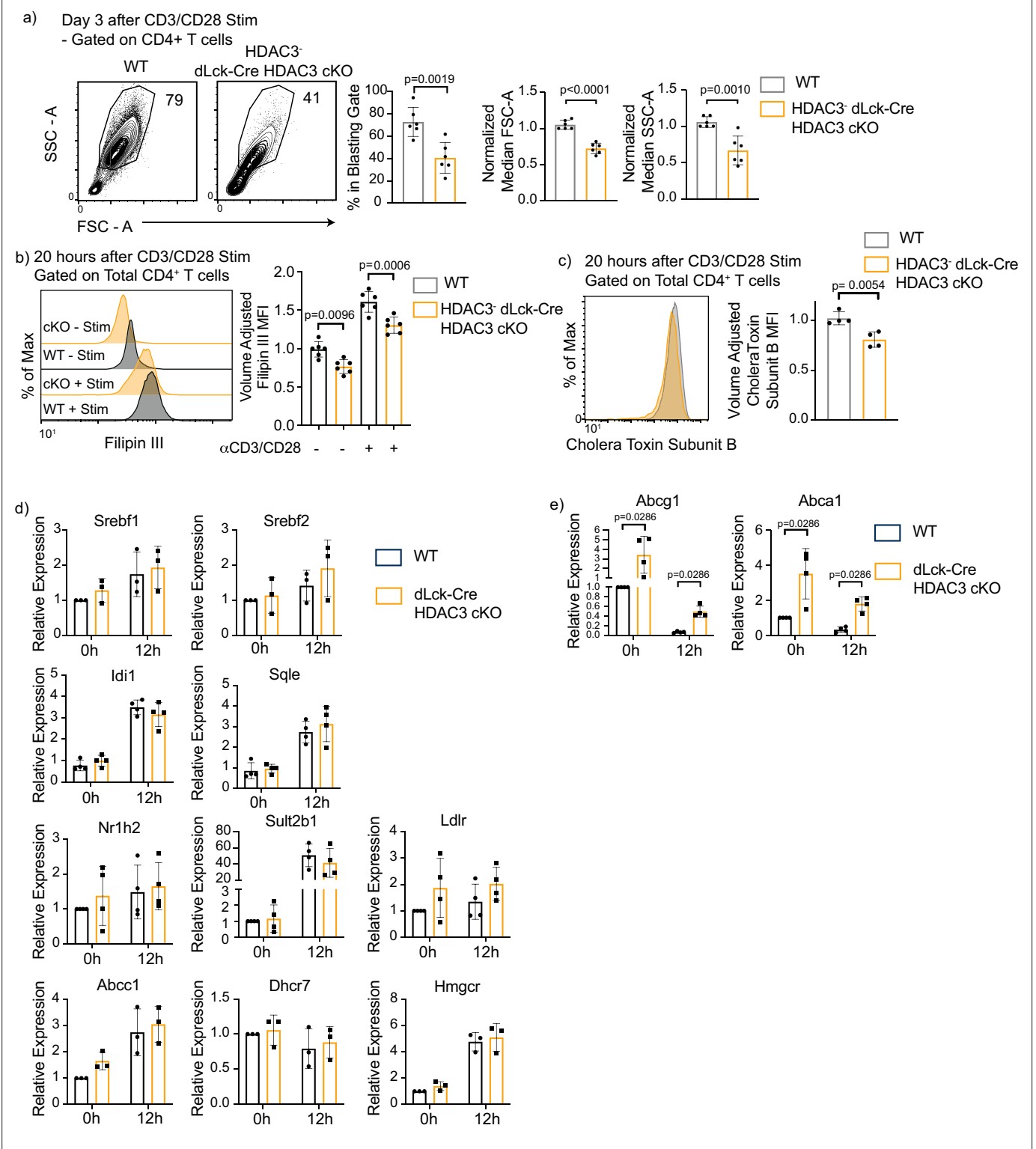

**Figure 5.** Histone deacetylase 3 (HDAC3)-deficient CD4+ T cells have defective blasting, reduced cholesterol levels, and increased cholesterol efflux transporter expression. (**a**) Enriched CD4+ T cells were stimulated with 5 µg plate-bound αCD3 and 1 µg/ml soluble αCD28 and analyzed for blasting by size (forward scatter area, FSC-A) and granularity (side scatter area, SSC-A) after 3 days. Bar graphs on right represent mean ± standard deviation (SD; *n* = 6 mice/group from three independent experiments). One nonlittermate, but age- and sex-matched wild-type (WT) B6 control was used in this experiment. Statistical significance was determined for the indicated comparisons using an unpaired *t*-test. (**b**) Splenocytes were isolated from WT or dLck-Cre HDAC3 cKO mice, and labeled for cholesterol using Filipin III 20 hr after stimulation with 5 µg of plate-bound αCD3 and 1 µg/ml of αCD28.

*Figure 5 continued on next page*

*Figure 5 continued*

Flow cytometry was conducted. Bar plot (right) shows volume-adjusted MFI ± SD quantified across three independent experiments (*n* = 6 mice/group). MFI was adjusted to approximate cell volume by taking MFI divided by FSC-W$^3$. Statistical significance was determined for the indicated comparisons using one-way analysis of variance (ANOVA) with Tukey's multiple comparisons test. (**c**) Enriched CD4$^+$ T cells were labeled with AF488-conjugated Cholera Toxin Subunit B 20 hr after stimulation with 5 µg of plate-bound αCD3 and 1 µg/ml of αCD28. Bar graph on right represents mean ± SD (*n* = 4 mice/group from two independent experiments). MFI was adjusted to approximate cell volume by taking MFI divided by FSC-W$^3$. Statistical significance was determined for the indicated comparison using an unpaired *t*-test. (**d, e**) Enriched CD4$^+$ T cells from Rag1-GFP WT and Rag1-GFP dLck-Cre HDAC3 cKO mice were sorted by fluorescence-activated cell sorting (FACS) for mature naive T cells (GFP$^-$). After sorting, cells were stimulated with 5 µg of plate-bound αCD3 and 1 µg/ml of αCD28, and RT-qPCR was conducted to examine gene expression of genes involved in cholesterol homeostasis. Bar graphs represent mean ± SD (*n* = 3–4 mice/group from three independent experiments). Statistical significance was determined for the indicated comparisons using an unpaired *t*-test.

Since cholesterol levels were disrupted in HDAC3-deficient CD4$^+$ T cells, key genes involved in cholesterol biosynthesis were examined. To enrich for HDAC3$^-$ MNTs without intracellular staining of HDAC3, dLck-Cre HDAC3 cKO mice were interbred to Rag1-GFP reporter mice. These mice have a knock in of GFP into the Rag1 locus (**Kuwata et al., 1999**). The half-life of GFP has been estimated to be ~56 hr in in vivo (**McCaughtry et al., 2007**). This stability allows GFP$^+$ T cells to be detected 2–3 weeks after T cell egress from the thymus even though Rag1 transcription ceases before T cells leave the thymus. Thus, Rag1-GFP$^-$ naive CD4$^+$ T cells are MNTs, and HDAC3$^-$ T cells comprise 75% of the MNT pool in dLck-Cre HDAC3 cKO (**Figure 1c**). Splenocytes from Rag1-GFP WT and Rag1-GFP dLck-Cre HDAC3 cKO mice were sorted using fluorescently activated cell sorting (FACS) for Rag1-GFP$^-$ mature naive CD4$^+$ T cells and cultured with or without αCD3/αCD28 antibodies for 12 hr to measure changes in gene expression. *Srebf1* and *Srebf2,* genes that produce sterol response element-binding proteins and serve as key regulators of lipid synthesis (**Bertolio et al., 2019**) showed similar expression between WT and HDAC3 cKO T cells before or after stimulation (**Figure 5d**). In addition, key genes involved in cholesterol synthesis and homeostasis such as *Hmgcr*, *Dhcr7*, *Sult2b1*, *Abcc1*, *Idi1*, and *Sqle* were all expressed normally in dLck-HDAC3 cKO cells (**Figure 5d**). Additionally, expression of *Nr1h2*, which encodes the protein Liver X Receptor Beta (LXRβ) and serves as a cholesterol sensitive transcription factor, also showed normal expression in dLck-Cre HDAC3 cKO T cells (**Figure 5d**). This finding is unexpected, as previous studies showed that disruptions in mTORC1 activity altered the expression of cholesterol synthesis genes (**Zeng et al., 2013**). However, HDAC3-deficient CD4$^+$ T cells had a higher Filipin III signal after TCR stimulation (**Figure 5b**), indicating they were still capable of increasing cholesterol synthesis despite reduced mTORC1 activation. Collectively, these data indicate TCR/CD28 signaling in HDAC3-deficient CD4$^+$ T cells is sufficient to drive the expression of cholesterol biosynthesis genes.

Since cholesterol levels were reduced despite intact cholesterol synthesis, changes in sterol export could be responsible for the reduced Filipin III signal in HDAC3-deficient CD4$^+$ T cells. To test this, gene expression of two ATP-binding cassette transporters, ABCA1 and ABCG1, was measured. ABCA1 and ABCG1 are pivotal players in cholesterol efflux (**Tarling and Edwards, 2012**). ABCG1 in particular has been identified as a key cholesterol transporter downregulated after T-cell activation and important for LXR's antiproliferative effects (**Bensinger et al., 2008**). At baseline, mature naive CD4$^+$ T cells from dLck-Cre HDAC3 cKO mice had a 3.4-fold increase in expression of *Abcg1* transcripts and a 3.5-fold increase in *Abca1* transcripts compared to WT (**Figure 5e**). Twelve hours after activation, *Abcg1* transcript expression in dLck-Cre HDAC3 cKO mice was 8.1-fold higher than WT activated T cells, while *Abca1* transcript expression in the activated dLck-Cre HDAC3 cKO was 7.1-fold higher than WT activated T cells (**Figure 5e**). *Abcg1* and *Abca1* expression was decreased with TCR stimulation in HDAC3-deficient cells after activation, but not to the level that occurred in WT CD4$^+$ T cells.

## Defective proliferation and blasting after activation of HDAC3-deficient CD4$^+$ T cells is rescued by addition of exogenous cholesterol in vitro

The decrease in cholesterol concentration in HDAC3-deficient CD4$^+$ T cells could drive inhibition of T-cell proliferation and blasting. Studies have shown that the addition of cholesterol conjugated to methyl-β-cyclodextrin (MBCD-Chol) can rescue CD8$^+$ T cells with defects in lipid homeostasis (**Kidani et al., 2013**). To test whether the addition of exogenous cholesterol improves HDAC3-deficient CD4$^+$ T-cell proliferation, CD4$^+$ T cells from WT and dLck-Cre HDAC3 cKO mice were labeled with CFSE and

cultured with αCD3/αCD28 simulation in the presence or absence of 5 µg/ml MBCD-Chol for 3 days. Remarkably, HDAC3-deficient CD4+ T cells that received exogenous cholesterol proliferated equivalently to WT T cells (*Figure 6a*). Additionally, HDAC3− dLck-Cre HDAC3 cKO CD4+ T cells treated with cholesterol had a strong recovery in the percentage of blasting cells compared to WT (*Figure 6b*). Importantly, cell size measured by FSC-A recovered to levels equivalent to WT (*Figure 6b*). However, cell granularity as measured by SSC-A did not return to WT levels in the cholesterol-treated HDAC3 cKO cells (*Figure 6b*), indicating HDAC3 may play a role in other pathways downstream of TCR signaling. WT CD4+ T cells that received exogenous cholesterol also increased their blasting and proliferation (*Figure 6a, b*), reinforcing the idea that cholesterol regulates the rate of CD4+ T-cell proliferation. These data confirmed that the reduction in cholesterol availability drove the defect in proliferation and blasting in the HDAC3-deficient CD4+ T cells. Interestingly, concentration of lipid rafts remained reduced in cholesterol-treated HDAC3− T cells (*Figure 6c*). Filipin III signal, as expected, was increased in the cholesterol-treated HDAC3− T cells (*Figure 6d*). Since lipid rafts remained disrupted in dLck-Cre HDAC3 cKO CD4+ T cells, we next asked whether the addition of cholesterol altered signaling events in HDAC3-deficient T cells. CD25 expression 3 days after in vitro stimulation was similar to WT levels in HDAC3-deficient T cells given exogenous cholesterol (*Figure 6e*). Conversely, p-S6 and Egr2 remained disrupted 16 hr after αCD3/αCD28 stimulation in the presence of exogenous cholesterol (*Figure 6f, g*). These data reveal that although mTORC1 signaling remained reduced in HDAC3-deficient CD4+ T cells, the residual signal was sufficient to drive normal proliferation given exogenous cholesterol. The ability of exogenous cholesterol to restore proliferation of HDAC3-deficient CD4+ T cells demonstrated that the primary function of HDAC3 is to repress cholesterol efflux during T-cell activation.

## HDAC3 loss results in increased expression and hyperacetylation of *Abca1* and *Abcg1* genes

HDAC3 represses genes through removal of acetyl groups from histone tails and other nonhistone proteins. Considering HDAC3-deficient CD4+ T cells have increased expression of ABCA1 and ABCG1, the *Abca1* and *Abcg1* gene loci were examined for changes in histone acetylation in the absence of HDAC3. Our group previously conducted both RNA-Seq and ChIP-Seq on FACS-sorted selecting (Vβ5intCD69+) thymocytes in OT-II WT and OT-II CD2-iCre HDAC3 cKO mice (*Philips et al., 2019a*). These data were examined to identify potential changes in transcript levels as well as histone acetylation at the gene loci for *Abca1* and *Abcg1*. HDAC3 cKO thymocytes had a significant upregulation of mRNA expression of *Abca1* and *Abcg1*, while *Hmgcr* expression was unaffected by HDAC3 loss (*Figure 7a*). This mirrored the expression data in the dLck-Cre HDAC3 cKO in peripheral CD4+ T cells (*Figure 5e*). Next, H3K9 and H3K27 acetylation at each of the gene loci for *Abca1*, *Abcg1*, and *Hmgcr* was examined. HDAC3 cKO thymocytes had a substantial increase of both H3K27 and H3K9 histone acetylation at the promoter regions of *Abca1* and *Abcg1* (*Figure 7b*). Previously, HDAC3 ChIP-Seq was conducted on human CD4+ T cells (*Wang et al., 2009*). These data were reexamined for HDAC3 enrichment at the *Abca1* and *Abcg1* loci. HDAC3 was enriched at five sites within the *Abcg1* locus, and HDAC3 was enriched at one site ~50 kilobases upstream of the *Abca1* locus (*Figure 7— figure supplement 1*). To further investigate whether the enzymatic activity of HDAC3 was required to suppress expression of these cholesterol efflux transporters in thymocytes, the 16610D9 thymocyte cell line was treated for 24 or 48 hr with RGFP966, a competitive tight-binding inhibitor of HDAC3, and mRNA expression of *Abca1*, *Abcg1*, and *Hmgcr* was measured after treatment. HDAC3 inhibition produced a profound upregulation of both *Abca1* and *Abcg1* expression after both 24 and 48 hr of HDAC3 inhibition (*Figure 7c*). Similarly, treatment of WT CD4+ T cells for 24 hr with RGFP966 significantly upregulated both *Abca1* and *Abcg1* expression in primary CD4+ T cells (*Figure 7d*). These experiments demonstrated the deacetlyase activity of HDAC3 was required to repress expression of *Abca1* and *Abcg1* expression during primary CD4+ T-cell activation.

## Discussion

In this study, we utilized a dLck-Cre HDAC3 cKO mouse system to examine the role of HDAC3 in peripheral CD4+ T cells. HDAC3-deficient CD4+ T cells have a defect in blasting and proliferation after in vitro αCD3/αCD28 stimulation. This defect is dependent upon reduced intracellular cholesterol

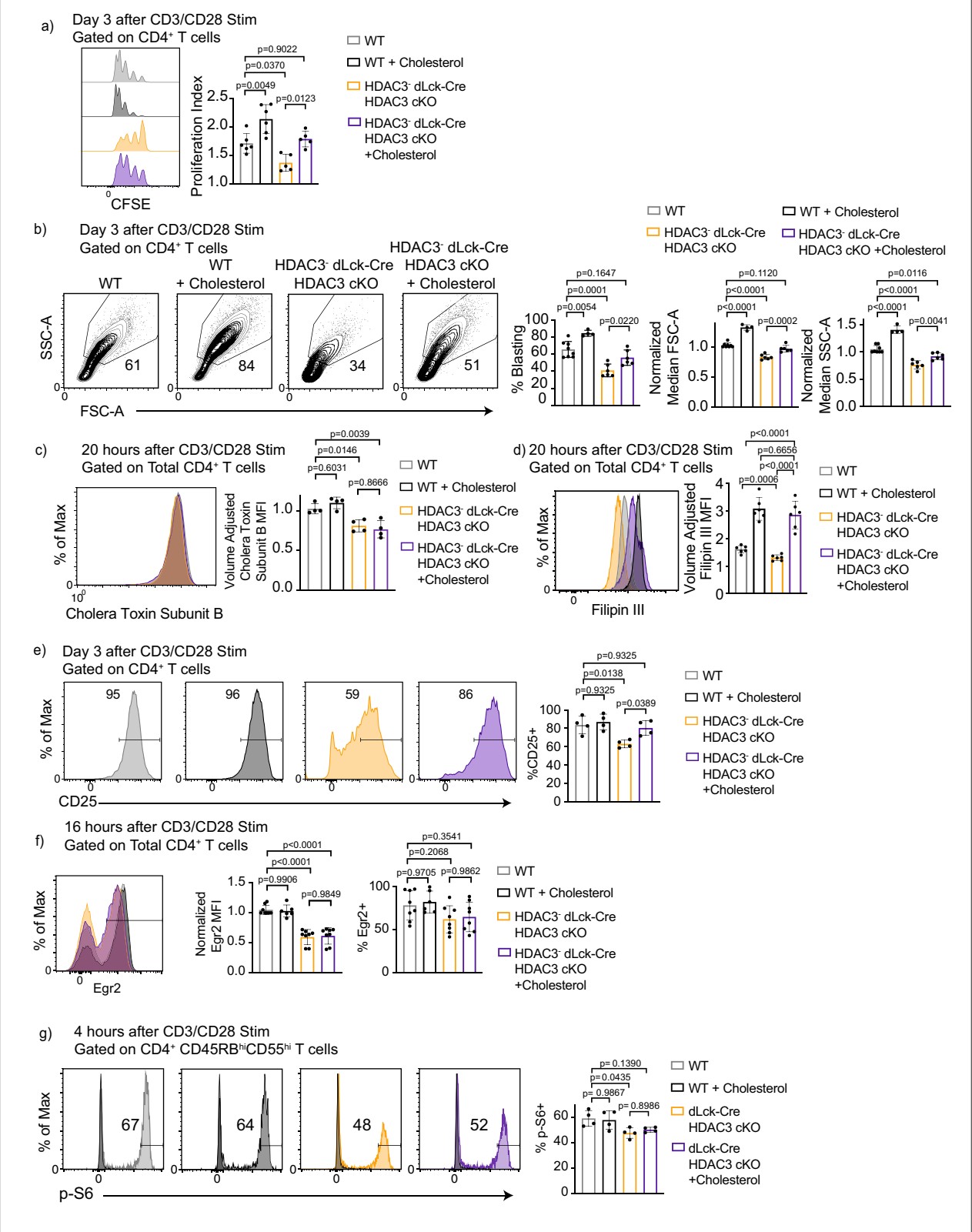

**Figure 6.** Defective proliferation and blasting in histone deacetylase 3 (HDAC3)-deficient CD4⁺ T cells is rescued by addition of exogenous cholesterol in vitro. (**a**) Enriched CD4⁺ T cells were labeled with CFSE and stimulated with 5 µg of plate-bound αCD3 and 1 µg/ml of αCD28 in the presence or absence of 5 µg cholesterol-methyl-β-cyclodextrin for 3 days and examined by flow cytometry for proliferation. Bar graphs represent mean proliferation index ± standard deviation (SD; *n* = 5 mice/group from three independent experiments). (**b**) Splenocytes were isolated from wild-type (WT) or dLck-

*Figure 6 continued on next page*

*Figure 6 continued*

Cre HDAC3 cKO mice, and magnetically enriched for CD4+ T cells by magnetic negative selection. Enriched CD4+ T cells were stimulated with 5 µg plate-bound αCD3 and 1 µg/ml soluble αCD28 in the presence or absence of 5 µg cholesterol-methyl-β- cyclodextrin, and analyzed for blasting by size (forward scatter area, FSC-A) and granularity (side scatter area, SSC-A) after 3 days. Bar graphs represent mean ± SD (*n* = 6–8 mice/group from three independent experiments). (**c**) Enriched CD4+ T cells were labeled with AF488-conjugated Cholera Toxin Subunit B 20 hr after stimulation with 5 µg of plate-bound αCD3 and 1 µg/ml of αCD28 in the presence or absence of 5 µg cholesterol-methyl-β-cyclodextrin. Bar graph on right represents mean ± SD (*n* = 4 mice/group from two independent experiments). MFI was adjusted to approximate cell volume by taking MFI divided by FSC-W³. (**d**) Enriched CD4+ T cells were labeled with Filipin III 20 hr after stimulation with 5 µg of plate-bound αCD3 and 1 µg/ml of αCD28 in the presence or absence of 5 µg cholesterol-methyl-β-cyclodextrin. MFI was adjusted to approximate cell volume by taking MFI divided by FSC-W³. Bar graph on right represents adjusted MFI ± SD (*n* = 6 mice/group from three independent experiments). (**e**) Enriched CD4+ T cells were stimulated with 5 µg plate-bound αCD3 and 1 µg/ml soluble αCD28 in the presence or absence of 5 µg cholesterol-methyl-β-cyclodextrin and analyzed for CD25 receptor expression after 3 days. Bar graphs represent mean percent CD25+ ± SD (*n* = 4 mice/group from two independent experiments). (**f**) Enriched CD4+ T cells were stimulated with 5 µg plate-bound αCD3 and 1 µg/ml soluble αCD28 in the presence or absence of 5 µg cholesterol-methyl-β-cyclodextrin and analyzed for Early Growth Response Protein 2 (Egr2) expression after 16 hr. Bar graphs represent percent Egr2+ ± SD and normalized Egr2 MFI ± SD among Egr2+ population (*n* = 8 mice/group from four independent experiments). (**g**) Enriched CD4+ T cells were stimulated with 5 µg plate-bound αCD3 and 1 µg/ml soluble αCD28 in the presence or absence of 5 µg cholesterol-methyl-β-cyclodextrin and analyzed phosphorylation of S6 after 4 hr. Bar graphs represent percent p-S6+ ± SD (*n* = 4 mice/group from two independent experiments). Statistical significance was determined for all indicated comparisons in this figure using a one-way analysis of variance (ANOVA) with Tukey's multiple comparisons test.

levels as addition of exogenous cholesterol in culture restores proliferation and blasting in HDAC3-deficient CD4+ T cells. Importantly, the addition of cholesterol does not restore diminished mTORC1 signaling in HDAC3− T cells, which implies that reduced mTORC1 in HDAC3 cKO T cells is sufficient to drive proliferation. Moreover, we have outlined a role for HDAC3 in the regulation of cholesterol efflux. In the absence of HDAC3, both resting and activated CD4+ T cells have increased expression of the cholesterol efflux transporters ABCA1 and ABCG1. Inhibition of HDAC3 in WT CD4+ T cells revealed that repression of *Abca1* and *Abcg1* was dependent upon HDAC3 enzymatic activity. Previous work has shown that components of the SMRT/NCOR complex play a critical role in regulating gene expression of cholesterol efflux transporter ABCG1 in macrophages (*Jakobsson et al., 2009*). However, the functional importance of the regulation of ABCG1 was not elucidated. Chromatin immunoprecipitation assays showed that HDAC3 localized to the promoter region of both ABCA1 and ABCG1 in human macrophages (*Jakobsson et al., 2009*). Further, HDAC3 ChIP-Seq in human CD4+ T cells revealed HDAC3 binding at the *Abcg1* locus, and upstream of the *Abca1* locus. In this study, we show that HDAC3-deficient thymocytes had hyperacetylation of the promoter region of the *Abca1* and *Abcg1* genes. In addition, we show that HDAC3 enzymatic inhibition by RGP966 led to increased ABCA1 and ABCG1 expression, demonstrating a critical role for HDAC3-mediated histone deacetylation in active suppression of ABCA1 and ABCG1. Thus, HDAC3 is a direct regulator of the *Abca1* and *Abcg1* genes in CD4+ T cells. Collectively, these data demonstrate that HDAC3 regulates cholesterol availability in CD4+ T cells and that HDAC3 is required during T-cell activation to transcriptionally repress cholesterol efflux through ABCA1 and ABCG1. Importantly, the regulation of cholesterol availability was the limiting factor blocking proliferation of HDAC3-deficient CD4+ T cells, as proliferation was restored by the addition of exogenous cholesterol.

T cells are uniquely sensitive to changes in cholesterol concentration shortly after activation, as recently activated T cells rapidly upregulate cholesterol synthesis and simultaneously abolish cholesterol efflux in preparation for the membrane production required for proliferation. Thus, disruptions in cholesterol synthesis produce profound defects in T-cell proliferation (*Kidani et al., 2013*). Likewise, it has been shown that enforced expression of the cholesterol efflux transporter ABCG1 through treatment with LXR agonists results in defective T-cell proliferation (*Bensinger et al., 2008*). Conversely, loss of LXRβ activity through genetic deletion alters T-cell fitness in the context of activation (*Michaels et al., 2021*), while genetic deletion of *Abcg1* in CD4+ T cells results in increased proliferation (*Armstrong et al., 2010*). Collectively, this demonstrates that transcriptional control of genes involved in cholesterol availability is critical for successful T-cell responses, and here we demonstrate a role for HDAC3 in controlling cholesterol availability in activated CD4+ T cells. This work demonstrates a highly specific role for HDAC3 in suppressing cholesterol efflux during T-cell activation. Importantly, critical components of the cholesterol synthesis pathway are unaltered by HDAC3 loss, demonstrating that HDAC3 serves as a specific regulator, rather than a generic regulator, of gene expression.

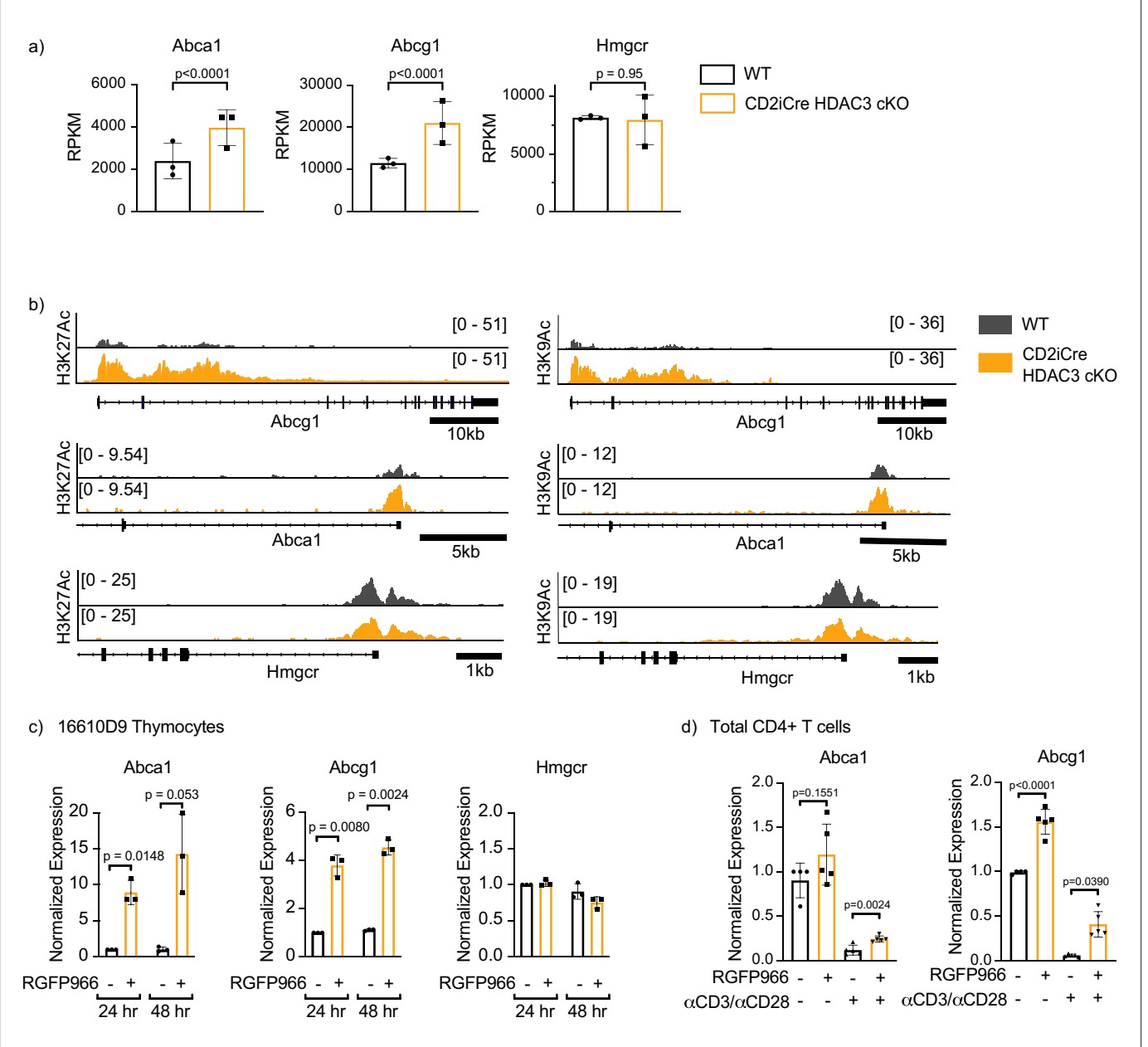

**Figure 7.** Histone deacetylase 3 (HDAC3) loss results in increased expression and hyperacetylation of ABCA1 and ABCG1. (**a**) Gene expression (RNA-Seq) of *Abca1*, *Abcg1*, and *Hmgcr* in selecting (TCRβ[int] CD69[+]) thymocytes from OT-II and OT-II CD2-iCre HDAC3 cKO mice. Bar graphs show mean RPKM (reads per kilobase million) ± standard deviation (SD). The exactTest (edgeR software) was used to compare mRNA levels (RPKM) from individual genes in RNA-Seq datasets. (**b**) Snapshot of H3K27Ac (left) and H3K9Ac (right) ChIP-seq tracks for the *Abca1*, *Abcg1*, and *Hmgcr* loci in selecting thymocytes from OT-II (wild-type, WT) and OT-II CD2-iCre HDAC3 cKO (CD2-iCre HDAC3-cKO) mice. (**c**) 16610D9 thymocytes were treated with competitive HDAC3 inhibitor RGFP966 for 24 or 48 hr, and expression of *Abca1*, *Abcg1*, and *Hmgcr* was measured by RT-qPCR at timepoints. Bar graphs show mean expression ± SD (*n* = 3 from three independent experiments). Statistical significance was determined for all indicated comparisons using an unpaired *t*-test. (**d**) WT splenocytes were harvested and magnetically enriched for CD4[+] T cells. Cells were cultured with 5 μg of plate-bound αCD3 and 1 μg/ml of αCD28 or left unstimulated. Cells were also treated with competitive inhibitor of HDAC3 RGFP966. After 24 hr, expression of Abca1 and Abcg1 was examined by RT-qPCR. Bar graphs show mean expression ± SD (*n* = 4–5 mice/group from three independent experiments).

The online version of this article includes the following figure supplement(s) for figure 7:

**Figure supplement 1.** Histone deacetylase 3 (HDAC3) binds to *Abcg1* gene locus, upstream of the *Abca1* locus in human CD4[+] T cells.

## Materials and methods

### Mice

*Hdac3 fl/fl* mice were generously provided by Dr. Scott Hiebert (*Knutson et al., 2008*). dLck-Cre mice were purchased from The Jackson Laboratory. Rag1-GFP knock-in mice were generously provided by Dr. Nobuo Sakaguchi (*Kuwata et al., 1999*). Mice were kept in barrier facilities and experiments were conducted with approval from the Institutional Animal Care and Use Committee at Mayo Clinic. Mice were analyzed between the age of 5–12 weeks, and both males and females were used. dLck-Cre HDAC3 cKO mice were examined with age-matched controls or littermates. WT mice may represent mice that have the floxed allele (*Hdac3 fl/fl*) alone, or mice that had no genetic alteration. The three instances of nonlittermate controls are noted in the figure legends. Sample size in the figure legends represents individual mice. Mouse genotypes were verified by flow cytometry analysis for HDAC3 protein expression or by PCR after use.

### Flow cytometry

Flow analysis was performed on the Attune NxT flow cytometer (Thermo Fisher) or the ZE5 Cell Analyzer (Bio-Rad), and all experiments were analyzed using FlowJo software (v10.5.3). To conduct intracellular flow cytometry, lymphocytes from spleen, thymus, or lymph nodes were labeled with surface markers, and then fixed with Foxp3/Transcription Factor Staining Buffer (eBioscience and Tonbo Biosciences). For analysis of phosphorylated protein analysis, the BD Phosflow kit was used. All flow analyses included size exclusion (FSC/SSC), doublet exclusion (FSC height/FSC area), and dead cell exclusion (Ghost Dyes; Tonbo Biosciences). Antibodies were purchased from eBioscience, BioLegend, Tonbo Biosciences, BD Biosciences, and Cell Signaling. Cells were labeled with Filipin III (Sigma #F4767) for 60 min in PBS at room temperature after fixation with Foxp3/Transcription Factor Staining Buffer. CTB (Thermo #34775) labeling was conducted for 30 min after Foxp3/Transcription Factor Staining Buffer.

### Magnetic enrichment of CD4$^+$ T cells

For stimulation and FACS sorting, cells were magnetically enriched using the EasySep Mouse Strepta-vadin RapidSpheres Isolation Kit (Stem Cell Technologies #19860) to remove non-CD4$^+$ T-cell populations from total splenocytes. Biotin-conjugated antibodies against CD8 (53–6.7), TCRγδ (UC7-13D5), NK1.1 (PK136), B220 (RA3-6B2), CD11b (M1/70), CD19 (6D5), CD11c (N418), Gr-1 (RB6-8C5), and Ter-119 (TER-119) were used. CD44 (IM7) was additionally used for negative selection to isolate naive CD4$^+$ T cells.

### FACS sorting

Cell sorting for qPCR was performed using the BD FACSMelody Cell Sorter. Magnetically enriched CD4$^+$ T cells from spleens of Rag1-GFP WT and RagGFP dLck-Cre HDAC3 cKO mice were labeled with anti-CD4 (RM4-5), anti-CD62L (MEL-14), and-CD44 (IM7), and Ghost Viability 510. Sorted MNTs were gated as live, CD4$^+$ CD62L$^+$ CD44$^-$ RagGFP$^-$ cells.

### Stimulations

To examine the expression of TCR signaling events, cells were stimulated with 5 µg/ml anti-CD3 (2C11, Bio X Cell #BE0001-1) and 1 µg/ml soluble anti-CD28 (37.51, Bio X Cell # BE0015-1). Tissue culture plates were coated with 5 µg/ml anti-CD3 in PBS for 3 hr at 37 °C. Splenocytes were magnet-ically enriched for CD4$^+$ T cells. Enriched cells were then cultured in complete culture media (RPMI 1640 with 10 % FCS, L-glutamine, penicillin and streptomycin, and β-mercaptoethanol) with stimula-tion. Unstimulated samples received 10 ng/ml IL-7 (PeproTech # 217-17) in complete media. Timing of each stimulation is noted in each figure legend. After stimulation, cells were immediately stained for flow cytometry on ice for 30 min, and fixed/permeabilized for intracellular staining as needed with the Tonbo Foxp3/Transcription Factor Staining Buffer Kit. For IL-2 detection, Protein Transport Inhibitor Cocktail (eBioscience # 00-4980-93) was added 6 hr prior to harvest. For antibodies targeting p-S6, stimulated cells were immediately fixed with BD Lyse/Fix Buffer (BD Phosflow kit), permeabilized with BD Perm Buffer III, and stained with anti-p-S6 antibodies (Cell Signaling #5364S) for 30 min at room

temperature. PE-conjugated αRabbit secondary antibody (Southern Biotech #4050-09) was used and stained for 15 min on ice.

## T-cell differentiation assays

For T-cell differentiation assays, cells were stimulated with 2 µg/ml plate-bound anti-CD3 (2C11) and 0.5 µg/ml soluble anti-CD28 (37.51) antibody. For $T_h1$ differentiation, complete culture media (RPMI 1640 with 10 % FCS, L-glutamine, penicillin and streptomycin, and β-mercaptoethanol) was supplemented with 1 µg/ml anti-IL-4 antibody (11B11, Bio X Cell #BE0045), 10 ng/ml IL-12 (PeproTech #210-12). For $T_h2$ differentiation, media was supplemented with 1 µg/ml of each anti-IFNγ (XMG1.2, Bio X Cell #BE0055) and anti-IL-12 antibody (R2-9A5, Bio X Cell BE0233) as well 10 ng/ml of IL-4 (PeproTech #214–14). For $T_h17$ differentiation, media was supplemented with 10 µg/ml of anti-IFNγ (XMG1.2, Bio X Cell #BE0055) and anti-IL-4 antibody (11B11, Bio X Cell #BE0045) as well as as 10 ng/ml of rIL-23 (PeproTech #200-23), 5 ng/ml TGF-β1 (PeproTech #100-21), and 20 ng/ml IL-6 (PeproTech 216-16). For $T_{reg}$ differentiation, media was supplemented with 10 µg/ml anti-IFNγ (XMG1.2, Bio X Cell #BE0055) and anti-IL-4 antibody (11B11, Bio X Cell #BE0045) as well as 2 ng/ml TGF-β1 PeproTech #100-21, and 2 ng/ml IL-2 (PeproTech #212-12). Cells were stimulated for 4 days, and stained for flow cytometry on ice. Unstimulated samples received 10 ng/ml IL-7 (PeproTech # 217-17) in complete media.

## CFSE labeling and proliferation assays

For proliferation assays, enriched CD4+ T cells were labeled with 1.25 µM CFSE (Sigma # 21888) for 2.5 min in PBS. Cells were washed with complete RPMI three times after labeling, and stimulated with 5 µg/ml plate-bound anti-CD3 (2C11) and 1 µg/ml soluble anti-CD28 (37.51) for 3–4 days. Cells were harvested and stained for flow cytometry on ice.

## Metabolic assays

The bioenergetic activity of CD4+ T cells was measured using the Seahorse XFe96 Analyzer. Magnetically enriched T cells were seeded at $2 \times 10^5$ cells/well. Cells were seeded on a Cell-Tak (Corning #354240)-coated XFe96 plate with fresh XF media (Seahorse XF RPMI medium with 2 mM glutamine, 10 mM glucose, 1 mM sodium pyruvate, and 5 mM HEPES (4-(2-hydroxyethyl)-1-piperazineethane sulfonic acid), pH 7.4). For the Mito Stress Assay, OCR was measured with additions of oligomycin (1.5 µM), FCCP (2-[2-[4-(trifluoromethoxy)phenyl]hydrazinylidene]-propanedinitrile, 1.5 µM), rotenone (1 µM), and antimycin A (1 µM) during the assay.

## Real-time quantitative PCR analysis

For qPCR analyses, mRNA was isolated from FACS sorted CD4+ T-cell populations or from 16610D9 thymocytes using Qiagen RNeasy Mini Kit. cDNA was generated using Superscript IV Reverse Transcriptase (Invitrogen). cDNA was amplified and detected using TaqMan probes for *Srebf1* (Thermo #Mm00550338_m1), *Srebf2* (Thermo #Mm01306292_m1), Idi1 (Thermo #Mm01337454_m1), *Sqle* (Thermo #Mm00436772_m1), *Nr1h2* (Thermo #Mm00437265_g1), *Sult2b1* (Thermo #Mm00450550_m1), *Ldlr* (Thermo #Mm00440169_m1), *Abcc1* (Thermo #Mm01344332_m1), *Dhcr7* (Thermo #Mm00514571_m1), *Hmgcr* (Thermo #Mm01282499_m1), *Abca1* (Thermo #Mm00442646_m1), *Abcg1* (Thermo #Mm00437390_m1) as well as an 18S rRNA (Applied Biosystems #4352930) to serve as an internal control. A StepOnePlus Real-Time PCR system was used, and differences in abundance were calculated using the 2-ΔΔCT method (*Livak and Schmittgen, 2001*).

## Cholesterol addition

Cholesterol-methyl-β-cyclodextrin (Sigma Prod# C4951) was added to complete RPMI at a concentration of 5 µg/ml (cholesterol weight). Stimulations and proliferation assays were performed as described above with or without cholesterol present in the media.

## ChIP-Seq and RNA-Seq

Chip-Seq and RNA-Seq data were previously published (*Philips et al., 2019a*; *Wang et al., 2009*). Data were retrieved from GEO Series GSE109531 and GSM393952. ChIP-Seq data were visualized with Integrated Genomics Viewer (mm10 for GSE109531, hg18 for GSM393952).

## Cell line

The 16610D9 murine double positive thymocyte cell line is used in *Figure 7c* only. This cell line was generated and received from Dr. Stephen Hedrick at UCSD. This cell line is not on the list of commonly misidentified cell lines and authentication for this cell line is not available from ATCS. To verify the identity of this cell line, flow cytometry using murine-specific antibodies for CD4, CD8, CD5, TCRβ, and CD24 was performed. The 16610D9 cells in this manuscript were CD4[+] CD8[+] double positive cells with intermediate expression of TCRβ, high expression of CD24, and low expression of CD5. This is consistent with the original description of the cell line. Mycoplasma testing was performed using the Universal Mycoplasma Detection Kit (ATCC Product # 30-1012 K), and the 16610D9 cell line was negative for mycoplasma contamination.

## Statistical analysis

The unpaired Student's *t*-test was used to compare between two groups for total cell counts, in vitro differentiation assays, proliferation, blasting, signal transduction flow cytometry, phosflow, filipin, and qPCR analysis. For comparisons with three or more groups, a one-way analysis of variance (ANOVA was used for total cell counts and HDAC3 expression in thymocytes, cell counts in splenocytes and mesenteric lymph nodes, proliferation, blasting and signaling events after cholesterol addition). The exactTest (edgeR software) was used to compare mRNA levels (RPKM) from individual genes in RNA-Seq datasets. *T*-tests and ANOVA analysis were calculated using GraphPad Prism. To calculate proliferation index in proliferation assays, the 'Generation 0' peak was set by drawing a gate around the unstimulated peak from an unstimulated sample in FlowJo. This gate was then applied to stimulated samples derived from the same mouse, and Flowjo calculated the proliferation index for each. For Filipin and CTB analysis, MFI was normalized to cell size by taking MFI divided by spheroid time of flight (FSC-W$^3$) to estimate cell volume as has been described previously (*Stein et al., 2017*; *Tzur et al., 2011*) Details for each statistical test used are included in each figure legend.

## Acknowledgements

We thank Dr. Scott Hiebert for *Hdac3*[flox] mice. We also thank Dr. Nobuo Sakaguchi for *Rag1-GFP* knock-in mice. We also thank members of the VSS, HZ, and Kay Medina (Mayo Clinic) laboratory for thoughtful discussions.

## Additional information

### Funding

| Funder | Grant reference number | Author |
| --- | --- | --- |
| National Institute of Allergy and Infectious Diseases | NIH R01 AI150100-01A1 | Virginia Smith Shapiro |
| National Institute of Allergy and Infectious Diseases | F31AI147438-01A1 | Drew Wilfahrt |
| A. Gary and Anita Klesch Predoctoral Fellowship | Mayo Clinic Graduate School of Biomedical Sciences | Drew Wilfahrt |

The funders had no role in study design, data collection, and interpretation, or the decision to submit the work for publication.

### Author contributions

Drew Wilfahrt, Conceptualization, Data curation, Formal analysis, Funding acquisition, Investigation, Validation, Writing – original draft, Validation, Visualization, Writing – original draft, Writing – review and editing; Rachael L Philips, Conceptualization, Data curation, Formal analysis, Funding acquisition, Investigation, Validation, Visualization, Writing – original draft, Writing – review and editing; Jyoti Lama, Monika Kizerwetter, Michael Jeremy Shapiro, Conceptualization, Data curation, Formal analysis, Investigation, Writing – review and editing; Shaylene A McCue, Conceptualization, Data curation,

Formal analysis, Investigation, Resources, Writing – review and editing; Madeleine M Kennedy, Matthew J Rajcula, Resources, Writing – review and editing; Hu Zeng, Conceptualization, Resources, Writing – review and editing; Virginia Smith Shapiro, Conceptualization, Formal analysis, Funding acquisition, Validation, Writing – original draft, Writing – review and editing

### Author ORCIDs
Monika Kizerwetter (iD) http://orcid.org/0000-0001-6084-7471
Virginia Smith Shapiro (iD) http://orcid.org/0000-0001-9978-341X

### Ethics
This study was performed in strict accordance with the recommendations in the Guide for the Care and Use of Laboratory Animals of the National Institutes of Health. All of the animals were handled according to approved institutional animal care and use committee (IACUC) protocols (#A00001984-16 and #A00004560-19) of the Mayo Clinic.

### Decision letter and Author response
Decision letter https://doi.org/10.7554/eLife.70978.sa1
Author response https://doi.org/10.7554/eLife.70978.sa2

---

## Additional files

### Supplementary files
• Transparent reporting form

### Data availability
All RNAseq and ChIP-seq data are publicly available in GEO (GSE109531 and GSE15735).

The following previously published datasets were used:

| Author(s) | Year | Dataset title | Dataset URL | Database and Identifier |
|---|---|---|---|---|
| Philips RL, Lee JH, Gaonkar K, Chanana P, Chung JY, Schwab A, Ordog T, Sinibaldo R, Arocha R, Shapiro VS | 2019 | HDAC3 restrains CD8-lineage genes to maintain a bi-potential state in CD4+CD8+ thymocytes for CD4-lineage commitment | https://www.ncbi.nlm.nih.gov/geo/query/acc.cgi?acc=GSE109531 | NCBI Gene Expression Omnibus, GSE109531 |
| Wang Z, Zang C, Cui K, Schones DE, Barski A, Peng W, Zhao K | 2009 | Genome-wide mapping of HATs and HDACs in human CD4+ T cells | https://www.ncbi.nlm.nih.gov/geo/query/acc.cgi?acc=GSE15735 | NCBI Gene Expression Omnibus, GSE15735 |

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

# Appendix 1

## Appendix 1—key resources table

| Reagent type (species) or resource | Designation | Source or reference | Identifiers | Additional information |
|---|---|---|---|---|
| Strain, strain background (*M. musculus*) | *Hdac3^flox* | Scott Hiebert | MGI: 379477 | PMID:18406327 |
| Strain, strain background (*M. musculus*) | dLck-Cre | Jackson Laboratory # 012837 | MGI: 4819511 | PMID:11748274 |
| Strain, strain background (*M. musculus*) | Cd4-Cre | Jackson Laboratory # 017336 | MGI: 2386448 RRID:IMSR_JAX:017336 | PMID:11728338 |
| Strain, strain background (*M. musculus*) | Rag1-GFP | Nobuo Sakaguchi | MGI: 2388344 | PMID:10586023 |
| Strain, strain background (*M. musculus*) | Cd2-iCre | Jackson Labs # 008520 | MGI: 2449947 RRID:IMSR_JAX:008520 | PMID:12548562 |
| Cell line (*M. musculus*) | 16610D9 | Stephen Hedrick | RRID:CVCL_0111 | PMID:10587351 |
| Antibody | Anti-CD4 PerCP (rat monoclonal) | Biolegend | Cat #100538 RRID:AB_893325 | Flow cytometry (1:500) |
| Antibody | Anti-CD4, BV510 (rat monoclonal) | Biolegend | Cat #100559 RRID:AB_2562608 | Flow cytometry (1:1000) |
| Antibody | Anti-CD55 af647 (Armenian hamster monoclonal) | Biolegend | Cat #131806 RRID:AB_1279261 | Flow cytometry (1:200) |
| Antibody | Anti-CD45RB Pacific Blue (rat monoclonal) | Biolegend | Cat #103316 RRID:AB_2174405 | Flow cytometry (1:200) |
| Antibody | Anti-CD8a BV510 (rat monoclonal) | Biolegend | Cat #100752 RRID:AB_2563057 | Flow cytometry (1:200) |
| Antibody | Anti-HDAC3 (rabbit monoclonal) | Cell Signaling | Cat #85057S RRID:AB_2800047 | Flow cytometry (1:1500) |
| Antibody | Anti-HDAC3 (mouse monoclonal) | Cell Signaling | Cat #3949S RRID:AB_2118371 | Flow cytometry (1:1000) |
| Antibody | Anti-TCRβ FITC (Armenian hamster monoclonal) | Biolegend | Cat #109206 RRID:AB_313429 | Flow cytometry (1:500) |
| Antibody | Anti-CD62L APC-Cy7 (rat monoclonal) | Tonbo Biosciences | Cat #25-0621U100 RRID:AB_2893432 | Flow cytometry (1:200) |
| Antibody | Anti-CD62L BV510 (rat monoclonal) | Biolegend | Cat #104441 RRID:AB_2561537 | Flow cytometry (1:200) |
| Antibody | Anti-CD44 BV510 (rat monoclonal) | Biolegend | Cat #103043 RRID:AB_2561391 | Flow cytometry (1:200) |
| Antibody | Anti-CD44 BV785 (rat monoclonal) | Biolegend | Cat #103041 RRID:AB_11218802 | Flow cytometry (1:200) |
| Antibody | Anti-CD44 v450 (rat monclonal) | Tonbo Biosciences | Cat #75-0441U025 RRID:AB_2621946 | Flow cytometry (1:200) |
| Antibody | Anti-Tbet PE-Cy7 (mouse monoclonal) | Biolegend | Cat #644824 RRID:AB_2561761 | Flow cytometry (1:100) |
| Antibody | Anti-GATA3 eFluor 660 (rat monoclonal) | eBioscience | Cat #50-9966-42 RRID:AB_10596663 | Flow cytometry (1:100) |
| Antibody | Anti-RORγt BV421 (mouse monoclona) | BD Horizon | Cat #562894 RRID:AB_2687545 | Flow cyotmetry (1:100) |
| Antibody | Anti-RORγt PE (rat monoclonal) | eBioscience | Cat #12-6981-82 RRID:AB_10807092 | Flow cytometry (1:100) |

*Appendix 1 Continued on next page*

*Appendix 1 Continued*

| Reagent type (species) or resource | Designation | Source or reference | Identifiers | Additional information |
|---|---|---|---|---|
| Antibody | Anti-Foxp3 Biotin (rat monoclonal) | eBioscience | Cat #13-5773-82 RRID:AB_763540 | Flow cytometry (1:100) |
| Antibody | Anti-Foxp3 PE (rat monclonal) | Tonbo Biosciences | Cat #50-5773U100 RRID:AB_11218868 | Flow cytometry (1:100) |
| Antibody | AAnti-CD25 PE-Cy7 (rat monoclonal) | Biolegend | Cat #102016 RRID:AB_312865 | Flow cytometry (1:500) |
| Antibody | Anti-PD-1 FITC (Armenian hamster monoclonal) | eBioscience | Cat #11-9985-85 RRID:AB_465473 | Flow cytometry (1:100) |
| Antibody | Anti-IL-4 BV421 (rat monoclonal) | Biolegend | Cat #504120 RRID:AB_2562102 | Flow cytometry (1:100) |
| Antibody | Anti-IFNγ FITC (rat monoclonal) | Invitrogen | Cat #RM9001 RRID:AB_10375014 | Flow cytometry (1:100) |
| Antibody | Anti-IL-17A Af488 (rat monoclonal) | Biolegend | Cat #506910 RRID:AB_536012 | Flow cytometry (1:100) |
| Antibody | Anti-CXCR5 BV421 (rat monoclonal) | Biolegend | Cat #145512 RRID:AB_2562128 | Flow cytometry (1:100) |
| Antibody | Anti-Bcl6 PE (mouse monoclonal) | Biolegend | Cat #648304 RRID:AB_2561375 | Flow cytometry (1:200) |
| Antibody | Anti-TNF-$\alpha$ PE (rat monoclonal) | Biolegend | Cat #506306 RRID:AB_315427 | Flow cytometry (1:100) |
| Antibody | Anti-IL-2 PE (rat monoclonal) | Biolegend | Cat #503808 RRID:AB_315302 | Flow cytometry (1:100) |
| Antibody | Anti-Egr2 PE (rat monoclonal) | eBioscience | Cat #12-6691-82 RRID:AB_10717804 | Flow cytometry (1:100) |
| Antibody | Anti-p-S6 (rabbit monoclonal) | Cell Signaling | Cat #5364S RRID:AB_10694233 | Flow cytometry (1:100) |
| Antibody | Anti-CD69 (Armenian hamster monoclonal) | Biolegend | Cat #104512 RRID:AB_493564 | Flow cytomery (1:200) |
| Antibody | Anti-P2RX7 (rabbit polyclonal) | Enzo Life Sciences | Cat #ALX-215-035R100 RRID:AB_2052434 | Flow cytometry (1:100) |
| Antibody | Anti-CD3 (Armenian hamster monoclonal) | Bio X Cell | Cat #BE0001-1 RRID:AB_1107634 | Stimulation of cultured cells (coated plate 3 hr at 37°C) |
| Antibody | Anti-CD28 (syrian hamster monoclonal) | Bio X Cell | Cat # BE0015-1 RRID:AB_1107624 | Stimulation of cultured cells (soluble) |
| Antibody | Anti IL-4 (rat monoclonal) | Bio X Cell | Cat #BE0045 RRID:AB_1107707 | Blocking for in vitro differentiation assays |
| Antibody | Anti-IFNγ (rat monoclonal) | Bio X Cell | Cat #BE0055 RRID:AB_1107694 | Blocking for in vitro differentiation assays |
| Antibody | Anti-IL-12 (rat monoclonal) | Bio X Cell | Cat #BE0233 AB_2687715 | Blocking for in vitro differentiation assays |
| Peptide, recombinant protein | TGF-β1 | PeproTech | Cat #100-21 | In vitro differentiation assays |
| Peptide, recombinant protein | IL-12 | PeproTech | Cat #210-12 | In vitro differentiation assays |
| Peptide, recombinant protein | IL-4 | PeproTech | Cat #214-14 | In vitro differentiation assays |
| Peptide, recombinant protein | IL-23 | PeproTech | Cat #200-23 | In vitro differentiation assays |

*Appendix 1 Continued on next page*

*Appendix 1 Continued*

| Reagent type (species) or resource | Designation | Source or reference | Identifiers | Additional information |
| --- | --- | --- | --- | --- |
| Peptide, recombinant protein | IL-6 | PeproTech | Cat #216-16 | In vitro differentiation assays |
| Peptide, recombinant protein | IL-2 | PeproTech | Cat #212-12 | In vitro differentiation assays |
| Chemical compound, drug | Methyl-β-cyclodextrin-cholesterol | Millipore Sigma | Cat #C4951 | |
| Chemical compound, drug | Filipin III | Millipore Sigma | Cat #F4767 | Flow cytometry (1:100) |
| Chemical compound, drug | Cholera Toxin Subunit B – Af488 | Thermo Fisher | Cat #C34775 | Flow cytometry (1:200) |
| Commercial assay or kit | BD Fix/Perm Buffer Kit (Phosflow) | BD Biosciences | Cat #558049; 558050 | |
| Commercial assay or kit | Foxp3/ Transcription Factor Staining Buffer Kit | Tonbo Biosciences | Cat #TNB-0607-KIT | |
| Chemical compound, drug | CFSE | Sigma | Cat #21888 | |
| Commercial assay or kit | EasySep Mouse Streptavadin Rapidspheres Isolation Kit | StemCell | Cat #19860 | |
| Antibody | Anti-CD8 Biotin (rat monoclonal) | Biolegend | Cat #100704 RRID:AB_312743 | Magnetic Enrichment – Negative Selection (1:100) |
| Antibody | Anti-TCRγδ Biotin (Armenian hamster monoclonal) | eBioscience | Cat #13-5811-85 RRID:AB_466685 | Magnetic Enrichment – Negative Selection (1:100) |
| Antibody | Anti-NK1.1 Biotin (mouse monoclonal) | Bioegend | Cat #13-5941-85 RRID:AB_466805 | Magnetic Enrichment – Negative Selection (1:100) |
| Antibody | Anti-B220 Biotin (rat monoclonal) | eBioscience | Cat #13-0452-85 RRID:AB_466450 | Magnetic Enrichment – Negative Selection (1:100) |
| Antibody | Anti-CD11b Biotin (rat monoclonal) | Bioegend | Cat #101207 RRID:AB_312787 | Magnetic Enrichment – Negative Selection (1:100) |
| Antibody | Anti-CD19 Biotin (rat monoclonal) | Bioegend | Cat #115503 RRID:AB_313638 | Magnetic Enrichment – Negative Selection (1:100) |
| Antibody | Anti-Cd11c Biotin (Armenian hamster monoclonal) | Bioegend | Cat #117303 RRID:AB_313772 | Magnetic Enrichment – Negative Selection (1:100) |
| Antibody | Anti-Gr-1 Biotin (rat monoclonal) | Bioegend | Cat #117303 RRID:AB_313368 | Magnetic Enrichment – Negative Selection (1:100) |
| Antibody | Anti-Ter-119 Biotin (rat monoclonal) | Bioegend | Cat #116203 RRID:AB_313704 | Magnetic Enrichment – Negative Selection (1:100) |
| Other | Fixable Viability Dye Ghost 510 | Tonbo Biosciences | Cat #13-0870 | Flow cytometry (1:1000) |

*Appendix 1 Continued on next page*

*Appendix 1 Continued*

| Reagent type (species) or resource | Designation | Source or reference | Identifiers | Additional information |
|---|---|---|---|---|
| Other | Fixable Viability Ghost 780 | Tonbo Biosciences | Cat #13-0865 | Flow cytometry (1:1000) |
| Other | Fixable Viability Dye Ghost 510 | Tonbo Biosciences | Cat #13-0870 | Flow cytometry (1:1000) |
| Commercial assay or kit | Seahorse XF Cell Mito Stress Test Kit | Agilent | Cat #103015-100 | |
| Sequence-based reagent | *Srebf1* | Thermo Fisher Scientific | #Mm00550338_m1 | |
| Sequence-based reagent | *Srebf2* | Thermo Fisher Scientific | #Mm01306292_m1 | |
| Sequence-based reagent | *Idi1* | Thermo Fisher Scientific | #Mm01337454_m1 | |
| Sequence-based reagent | *Sqle* | Thermo Fisher Scientific | #Mm00436772_m1 | |
| Sequence-based reagent | *Nr1h2* | Thermo Fisher Scientific | #Mm00437265_g1 | |
| Sequence-based reagent | *Sult2b1* | Thermo Fisher Scientific | #Mm00450550_m1 | |
| Sequence-based reagent | *Idlr* | Thermo Fisher Scientific | #Mm00440169_m1 | |
| Sequence-based reagent | *Abcc1* | Thermo Fisher Scientific | #Mm01344332_m1 | |
| Sequence-based reagent | *Dhcr7* | Thermo Fisher Scientific | #Mm00514571_m1 | |
| Sequence-based reagent | *Hmgcr* | Thermo Fisher Scientific | #Mm01282499_m1 | |
| Sequence-based reagent | *Abcg1* | Thermo Fisher Scientific | #Mm00437390_m1 | |
| Sequence-based reagent | *Abca1* | Thermo Fisher Scientific | #Mm00442646_m1 | |
| Sequence-based reagent | *hmgcr* | Thermo Fisher Scientific | Mm01282499_m1 | |
| Sequence-based reagent | *18*S | Applied Biosystems | Cat #4352930 | |
| Software, algorithm | Prism 8 | GraphPad | RRID:SCR_002798 | |
| Software, algorithm | FlowJo v10 | Treestar | RRID:SCR_008520 | |
| Software, algorithm | Illustrator | Adobe | RRID:SCR_010279 | |
| Software, algorithm | IGV | Broad Institute | RRID:SCR_011793 | |

