## [Editor Report]

This paper will be of interest to scientists in the field of T cell biology and immunometabolism. The data analysis is rigorous and the experiments performed are appropriate. The findings of the manuscript will expand upon previous findings of a role for histone deacetylase 3 in thymocyte development and CD8^+^ T cell function to that of CD4^+^ T cells.

---

## [Decision Letter]

**Decision letter after peer review:**

Thank you for submitting your article "Histone deacetylase 3 represses cholesterol efflux during CD4 + T cell activation" for consideration by *eLife*. Your article has been reviewed by 2 peer reviewers, and the evaluation has been overseen by a Reviewing Editor and Satyajit Rath as the Senior Editor. The reviewers have opted to remain anonymous.

Essential revisions:

Both reviewers provided detailed recommendations which are also provided below for clarity and context. Among these, it would be good to focus on the following:

1) The staining for T_fh_ cells in Figure 2 is not convincing. The shift of the Bcl6-positive population is rather low and therefore it is difficult to gate of T_fh_ cells. In addition, the number of T_fh_ cells for non-immunized mice seems to be rather high. This might be related to the gating strategy. It might be better to use PD1 vs CXCR5 for the definition of T_fh_ cells (and then gate on Bcl6). Did the authors analyze the frequency of T_fh_ cells in peyer's patches (in WT mice, T_fh_ cells are present in PP at steady state)?

2) Have the authors analyzed cytokine expression in ex vivo isolated CD4^+^ T cells (stimulated with PMA/ionomycin). It would be very informative to know whether T-bet+ or RORgt+ HDAC3- CD4^+^ T cells display changes in IFNγ or IL-17A expression, respectively.

3) The authors need to more convincingly demonstrate that the blasting/proliferation phenotype is related to the cholesterol content. There is some reservations about the filipin staining that can be easily addressed by looking at their existing data. It would be beneficial to see the cholesterol loading experiments redone where they aren't overloading the cells with cholesterol and have the proper control.

*Reviewer #1:*

The study "Histone deacetylase 3 represses cholesterol efflux during CD4^+^ T cell activation" represents a detailed study on the role of HDAC3 on CD4^+^ T cell activation. The reader is presented with detailed analysis on how loss of HDAC3 alters cholesterol metabolism in CD4^+^ T cells. The provided data are well presented and conclusive.

Overall, this is a nice study. Some comments are listed below.

(1) In Figure 1c, the data would be easier to read if the order of the statistical summary showing the percentage of the various cell populations correlates with the sequence of the representative FACS plots.

(2) In Figure 2 the authors should display data from the same organ (spleen) for all subsets, while mLN data should be presented exclusively in the corresponding supplementary Figure.

(3) The staining for T_fh_ cells in Figure 2 is not convincing. The shift of the Bcl6-positive population is rather low and therefore it is difficult to gate of T_fh_ cells. In addition, the number of T_fh_ cells for non-immunized mice seems to be rather high. This might be related to the gating strategy. It might be better to use PD1 vs CXCR5 for the definition of T_fh_ cells (and then gate on Bcl6). Did the authors analyze the frequency of T_fh_ cells in peyer's patches (in WT mice, T_fh_ cells are present in PP at steady state)?

(4) Have the authors analyzed cytokine expression in ex vivo isolated CD4^+^ T cells (stimulated with PMA/ionomycin). It would be very informative to know whether T-bet+ or RORgt+ HDAC3- CD4^+^ T cells display changes in IFNγ or IL-17A expression, respectively.

(5) It is not clear, at least to the reviewer, why the authors show the % of CD4^+^ T cells as well as total cell numbers from CD4-Cre HDAC3cKO mice in the diagrams in the right panels of Figure 2. If they include the data, one would also have to show the % and number of the corresponding WT cells from the study.

(6) Which cells were used as negative control in Figure 3 as well as in Figure 3—figure supplement 1. This is not described in the manuscript. Ideally, the negative controls are unboosted but stained cells. If the negative controls are unstained cells, the negative control peaks might in fact not correlate with the negative cytokine peak of the actual sample. The gating for IFNγ, IL-4 and IL-17 (is it IL-17A?) cytokine expression doesn't look okay, e.g. there are two peaks for IFNγ expression, but the gates include cells that appear IFNγ-negative. Similarly, the authors show 90% IL-17+ cells. This is very high, however, there seems to be a small fraction of cells that express very high levels. Are these cells the "real" IL-17A positive cells?

Showing contour plots (or dot plots) CD4 vs cytokine might help to discriminate between cytokine negative and positive peaks.

Have the cells been restimulated/boosted with PMA/ionomycin?

(7) Approx. 50% of naïve CD4^+^ T cells have deleted HDAC3 (as shown in Figure 1c). After differentiation into the various lineages shown in Figure 3, what was the % of cells that had deleted HDAC3. Was there a competitive advantage of cells that didn't delete HDAC3?

(8) The authors should briefly explain why they used Rag1-GFP mice for the data presented in Figure 5d and 5e and why the sorted GFP- cells for this experiment. Not all readers might be familiar with these mice.

(9) Related to Figure 6a: would the addition of cholesterol enhance the proliferation of WT CD4^+^ T cells?

(10) Why was a thymocyte cell line used for the data presented in Figure 7c? Wouldn't it be better to show results from primary cells?

*Reviewer #2:*

Past work of this lab and others have described a role for the histone deacetylase 3 (HDAC3) in the regulation of thymocyte development and CD8^+^ T cell function. This study used a loss-of-function approach to study the role of HDAC3 in CD4^+^ T cells. This was done by studying the T cell compartment in HDAC3 floxed mice that were crossed to mice expressing the distal Lck-Cre promoter (HDAC3 T KO mice). They observed that HDAC3 T KO mice exhibited CD8, but not CD4 T cell lymphopenia and a normal naïve to memory CD4^+^ T cell ratio convincingly establishing that Th cell development was intact in these mice. They then established using in vitro assays that CD4^+^ T cells had defects in blasting and proliferation that correlated with reduced expression IL-2 and CD25, and reduced expression and activity of mTORC1 targets. They then focused their analysis on cholesterol metabolism-related genes in the CD4^+^ T cells and discovered that two cholesterol efflux proteins were upregulated in HDAC3-deficient CD4^+^ T cells at the mRNA level. Past work in the literature had established that cholesterol efflux is downregulated during T cell activation to support T cell growth. They also showed that loading T cells with cholesterol restored the defect in T cell blasting and proliferation and CD25 expression in HDAC3 KO CD4^+^ T cells.

Strengths:

The manuscript was well-written and concisely presented. The experiments that were performed generated data of high quality. Experimental approaches were appropriate for surveying defects in the T cell compartment and effector T cell function. Experiments were replicated and, in most instances, means represented means of experiments, which is to be commended. Statistics were appropriate for each analysis that was performed.

With the exception of some of the data in Figure 6, the conclusions match the data presented.

Weaknesses:

The conclusion that "HDAC3-deficiency results in cholesterol deficiency and this is the cause of the proliferation defects" could be strengthened by the following considerations. First, since they gated on total CD4^+^ T cells, which were overall smaller in size in the HDAC3 KO mice, it would be important to show that the lowered filipin and cholera toxin staining is not a result of reduced autofluorescence.

In addition, the loading of the cells with cholesterol appeared to be almost too efficient and resulted in higher filipin staining than the WT T cells (Figure 6). To prove that the effects of the HDAC3-deficiency on T cell blasting and proliferation are indeed related to the cholesterol deficiency, the experiment should attempt to restore, not increase cholesterol levels. A cholesterol-loaded WT should also be included as a control. In addition, since cholesterol efflux is not being measured in the study, conclusions could have been strengthened by staining for cholesterol transporters using commercial antibodies.

In addition, the manuscript would have been improved if certain details were provided about the nature of the WT mice used in each experiment. The methods described that the littermate floxed mice were used as controls in some experiments and "off-the-shelf" C57BL6/J mice were used in others and the term WT was used interchangeably to describe these different mice. The legends also did not detail what the sample size represented (individual mice or cultures) and what sex of mice were used for each experiment.

Comments for the authors:

1. It is surprising that the filipin staining is lowered in T cells cultured in full serum media. I have conducted such studies in T cells cultured with HMG CoA reductase inhibitors and as unable to see decreases in filipin staining in T cells unless the T cells were cultured with HMG-CoA reductase inhibitors with serum-free media. This suggested to me that the media could be an important source of cholesterol for these cells. This aspect was not considered.

2. It would be good to gate the blasting and non-blasting cells with similar sized gates in the WT and KO T cells in the FSC/SSC plot before looking at filipin staining to rule out size effects on autofluorescence.

3. The finding that the deletion appears to be more efficient in the naïve mature T cells (25% still expressing), versus the memory T cells (75%), suggests that the HDAC3 defect may have given the remaining HDAC3+ T cells a selective advantage to enter the memory T cell pool as a result of homeostatic proliferation. This possibility was not considered.

4. Legends are sometimes confusing. For example, for Figure 4 (are these HDAC- gated cells from dLck-Cre mice?). If so, what is the WT control. Do you have the internal WT control (HDAC3+ cells in the dLck-Cre floxed mice)? Why was this shown for some experiments and not others.

5. I didn't find the rationale completely convincing to justify the focus on cholesterol metabolism, when IL-2 and IL-2R expression were decreased by more than 50%! mTORC and metabolic changes occur in part downstream of IL-2R (and CD28). IL-2 and IL-2R defects could have explained the phenotype.

---

## [Author Response]

Essential revisions:Both reviewers provided detailed recommendations which are also provided below for clarity and context. Among these, it would be good to focus on the following:1) The staining for T_fh_ cells in Figure 2 is not convincing. The shift of the Bcl6-positive population is rather low and therefore it is difficult to gate of T_fh_ cells. In addition, the number of T_fh_ cells for non-immunized mice seems to be rather high. This might be related to the gating strategy. It might be better to use PD1 vs CXCR5 for the definition of T_fh_ cells (and then gate on Bcl6). Did the authors analyze the frequency of T_fh_ cells in peyer's patches (in WT mice, T_fh_ cells are present in PP at steady state)?

We have altered the gating scheme to match what was proposed by the reviewer. Figure 2 and its associated supplements 1 and 2 now utilize the two-step strategy of gating on PD-1^+^ CXCR5^+^ cells and then gating on the Bcl-6^hi^ cells. We also added data for T_fh_ cells in the Peyer’s patch in Figure 2—figure supplement 2. Using this gating strategy, we see a significant reduction in total T_fh_ numbers in the spleen in the dLck-Cre HDAC3 cKO compared to WT. There was not a significant difference in T_fh_ frequency between dLck-Cre HDAC3 cKO and WT in the mLN or Peyer’s patches.

2) Have the authors analyzed cytokine expression in ex vivo isolated CD4^+^ T cells (stimulated with PMA/ionomycin). It would be very informative to know whether T-bet+ or RORgt+ HDAC3- CD4^+^ T cells display changes in IFNγ or IL-17A expression, respectively.

We have added this data into Figure 3 – supplement 2, and lines 171-178. Short-term stimulation worked well for IFN-γ in splenocytes, and was comparable between WT and HDAC3-deficient CD4^+^ Tbet^+^ T cells. This is consistent with our in vitro results (Figure 3—figure supplement 1) We conducted short-term (6 hour) stimulation of cells from mesenteric lymph node, where RORγt^+^ CD4^+^ T cells are more abundant, but we were unable to detect IL-17A production in either the WT or HDAC3 cKO cells after PMA/ionomycin stimulation.

3) The authors need to more convincingly demonstrate that the blasting/proliferation phenotype is related to the cholesterol content. There is some reservations about the filipin staining that can be easily addressed by looking at their existing data.

The HDAC3-deficient CD4 T cells have reduced cholesterol content (as measured by Filipin III signal) at all sizes compared to WT cells. This can be seen in Author response image 1 which shows FSC-A vs. Filipin III signal.

**Author response image 1. sa2fig1:** 

We have now normalized the Filipin III and Cholera Toxin Subunit B MFI to cell size in our data using a previously established method by taking MFI and dividing by an estimated volume of the cell using FSC-W^3^ (Stein et al., 2017; Tzur, Moore, Jorgensen, Shapiro, and Kirschner, 2011). The data in figure 5 still demonstrates a significant reduction of Filipin III signal and Cholera Toxin Subunit B signal in the HDAC3^-^ dLck-Cre HDAC3 cKO CD4^+^ T cells, confirming that HDAC3^-^ dLck-Cre HDAC3 cKO CD4^+^ T cells have a reduction in cholesterol content independent from changes in cellular volume during blasting and activation.

It would be beneficial to see the cholesterol loading experiments redone where they aren't overloading the cells with cholesterol and have the proper control.

The concentration of methyl-β-cyclodextrin cholesterol (MBCD-Chol) we used in the paper (5ug/mL) was the lowest biologically relevant concentration for restoration of proliferation. Titration experiments using lower concentrations had little to no effect on the function of dLck-Cre HDAC3-cKO or WT cells. We speculate that this may be related to the mechanism of MBCD-Chol incorporation into cells. Little is known as to whether MBCD-Chol incorporates into membranes or lipid rafts with the same efficiency or frequency into the various cellular compartments as unconjugated cholesterol. It is possible that although the cells are taking up a great deal of cholesterol (as shown by the increase in Filipin III staining), only a fraction of that is providing a physiological benefit to the cells. We have added in the WT+ Cholesterol control to all experiments in figure 6 to demonstrate the effects of exogenous cholesterol on wild type cells as a control.

Reviewer #1:The study "Histone deacetylase 3 represses cholesterol efflux during CD4^+^ T cell activation" represents a detailed study on the role of HDAC3 on CD4^+^ T cell activation. The reader is presented with detailed analysis on how loss of HDAC3 alters cholesterol metabolism in CD4^+^ T cells. The provided data are well presented and conclusive.Overall, this is a nice study. Some comments are listed below.(1) In Figure 1c, the data would be easier to read if the order of the statistical summary showing the percentage of the various cell populations correlates with the sequence of the representative FACS plots.

This has been corrected, the order of the FACS plots and bar graphs now match.

(2) In Figure 2 the authors should display data from the same organ (spleen) for all subsets, while mLN data should be presented exclusively in the corresponding supplementary Figure.

This has been changed with all spleen samples in Figure 2, all mLN in Figure 2 —figure supplement 1 and the updated Peyer’s patch addition in Figure 2 —figure supplement 2.

(3) The staining for T_fh_ cells in Figure 2 is not convincing. The shift of the Bcl6-positive population is rather low and therefore it is difficult to gate of T_fh_ cells. In addition, the number of T_fh_ cells for non-immunized mice seems to be rather high. This might be related to the gating strategy. It might be better to use PD1 vs CXCR5 for the definition of T_fh_ cells (and then gate on Bcl6).

Figure 2 and its supplements have been updated, please see the full response is in the “essential revisions 1”.

(4) Have the authors analyzed cytokine expression in ex vivo isolated CD4^+^ T cells (stimulated with PMA/ionomycin).

Figure for IFN-γ after short term PMA/Ionomycin stim has been added (Figure 3—figure supplement 2). Please see the full response is in “essential revisions 2”.

(5) It is not clear, at least to the reviewer, why the authors show the % of CD4^+^ T cells as well as total cell numbers from CD4-Cre HDAC3cKO mice in the diagrams in the right panels of Figure 2. If they include the data, one would also have to show the % and number of the corresponding WT cells from the study.

To simplify this figure, we removed the % of CD4^+^ T cell bar graphs from figure 2. For Figure 2 – supplement 1 we now only report the frequency of CD4^+^ T cells from the mLN as the number of lymph nodes harvested from each mouse was variable, the frequency of CD4^+^ T cells is a more appropriate comparator than total cell number. For clarity, we added in the flow analysis for the CD4-Cre HDAC3 cKO mice. The CD4-Cre HDAC3 cKO cells have a block in thymocyte development that results in very few mature T cells (and thus differentiated CD4^+^ T cells) in vivo*,* and thus serve as a good control for the loss of the differentiated CD4^+^ T cell numbers. Further, this loss of mature T cells in CD4-Cre HDAC3 cKO mice produces a relatively non-competitive environment, which allows for the development of relatively normal frequencies of differentiated T_reg_, T_h_2, and T_h_17 CD4 populations in the spleen (Figure 2), despite dramatically reduced total numbers. This reinforces the idea that HDAC3-deficient T cells are capable of differentiating, but are outcompeted by the WT cells in the competitive environment of the dLck-Cre HDAC3 cKO in vivo due to poor proliferative capacity.

(6) Which cells were used as negative control in Figure 3 as well as in Figure 3—figure supplement 1. This is not described in the manuscript. Ideally, the negative controls are unboosted but stained cells. If the negative controls are unstained cells, the negative control peaks might in fact not correlate with the negative cytokine peak of the actual sample.

We apologize for this oversight. This information has been added into the Figure legends of Figure 3 and Figure 3—figure supplement 1. The negative control (black histograms) is indeed unstimulated but stained cells as recommended by the reviewer.

The gating for IFNγ, IL-4 and IL-17 (is it IL-17A?) cytokine expression doesn't look okay, e.g. there are two peaks for IFNγ expression, but the gates include cells that appear IFNγ-negative. Similarly, the authors show 90% IL-17+ cells. This is very high, however, there seems to be a small fraction of cells that express very high levels. Are these cells the "real" IL-17A positive cells?

The stain is using an IL-17A antibody (Biolegend Cat #506910). We set a positive gate for all cytokine gates based on the unstimulated controls (black histograms figure 3 – supplement 1). As over 90% of the cells in our T_h_17 differentiation assays were RORγt^+^ (Figure 3), the percentage of IL-17A producing cells in this figure is consistent with this. Likewise, over 75% of the cells in the T_h_1 differentiation assays were T-bet^+^ (Figure 3), so the percentage of IFN-γ^+^ T cells is consistent with this.

Showing contour plots (or dot plots) CD4 vs cytokine might help to discriminate between cytokine negative and positive peaks.

We did not find any differences in the delineation of cytokine positive and negative peaks using either of these methods, and thus kept the original histogram plots shown.

Have the cells been restimulated/boosted with PMA/ionomycin?

These cells were not restimulated/boosted with PMA/ionomycin prior to harvest. They were, however, kept in the presence αCD3/CD28 stimulation throughout the duration of the in vitro differentiation cultures.

(7) Approx. 50% of naïve CD4^+^ T cells have deleted HDAC3 (as shown in Figure 1c). After differentiation into the various lineages shown in Figure 3, what was the % of cells that had deleted HDAC3. Was there a competitive advantage of cells that didn't delete HDAC3?

We didn’t find a significant enrichment of HDAC3^+^ cells from the dLck-Cre HDAC3 cKO mice in the differentiation assays. This result is likely confounded by the deletion timing of HDAC3 in the dLck-Cre HDAC3 cKO system. Most (~65-70%) of the recent thymic emigrants (RTEs) are HDAC3 sufficient, while very few (~20%) of the mature naïve T cells are HDAC3 sufficient (Figure 1c). Thus, most of the magnetically-enriched naïve HDAC3-sufficient cells are RTEs, which have been previously shown to have significant functional impairment compared to mature naïve T cells (Cunningham, Helm, and Fink, 2018).Thus, the naïve HDAC3-sufficient cells from the dLck-Cre HDAC3 cKO mice likely do not have a competitive advantage in the stimulation cultures since most of them are also functionally impaired RTEs.

(8) The authors should briefly explain why they used Rag1-GFP mice for the data presented in Figure 5d and 5e and why the sorted GFP- cells for this experiment. Not all readers might be familiar with these mice.

We apologize for this oversight. This has been added into the body of the manuscript within the Results section of figure 5 (lines 260-268).

(9) Related to Figure 6a: would the addition of cholesterol enhance the proliferation of WT CD4^+^ T cells?

We have added the “WT + Cholesterol” control to all panels in figure 6. WT cells readily take up the exogenous cholesterol (as measured by Filipin III), and this improves their proliferation as it likely bypasses their need for upregulation of production of cholesterol after activation.

(10) Why was a thymocyte cell line used for the data presented in Figure 7c? Wouldn't it be better to show results from primary cells?

We have added qPCR data from primary CD4 T cells treated with the competitive HDAC3 inhibitor, RGFP966, with and without the presence of CD3/CD28 stimulation. This data is in figure 7d.

Reviewer #2:[…] First, since they gated on total CD4^+^ T cells, which were overall smaller in size in the HDAC3 KO mice, it would be important to show that the lowered filipin and cholera toxin staining is not a result of reduced autofluorescence.

Our Filipin III and Cholera Toxin Subunit B analysis was at a 20hr stimulation timepoint, which should be before the cells have significantly begun to blast. Still, to account for any modest changes in cell size, we normalized the Filipin III and Cholera Toxin Subunit BMFI to cell volume. Detailed information on this volume adjustment can be seen in “essential revisions 3”.

In addition, the loading of the cells with cholesterol appeared to be almost too efficient and resulted in higher filipin staining than the WT T cells (Figure 6). To prove that the effects of the HDAC3-deficiency on T cell blasting and proliferation are indeed related to the cholesterol deficiency, the experiment should attempt to restore, not increase cholesterol levels.

Please see response in “Essential Revisions 4”.

A cholesterol-loaded WT should also be included as a control.

All panels in figure 6 now include a cholesterol loaded wild type as a control.

In addition, since cholesterol efflux is not being measured in the study, conclusions could have been strengthened by staining for cholesterol transporters using commercial antibodies.

We attempted to detect ABCA1 and ABCG1 proteins using commercially available antibodies by western blot on sorted CD4 T cells, but the antibodies we tried (Novus Biologicals #NB100-2068 for ABCA1, and Novus Biologicals# NB400-132 for ABCG1) failed to produce bands at the expected molecular weight by western blot in any of our WT or HDAC3 cKO samples. Given that much work has been done to show that both ABCA1 and ABCG1 protein levels are tightly controlled by LXR-dependent transcriptional events (Bensinger et al., 2008; Chen et al., 2011; Tan et al., 2017), we believe that the qPCR analysis in this manuscript demonstrates the role of HDAC3 in suppressing *Abca1* and *Abcg1* gene expression.

In addition, the manuscript would have been improved if certain details were provided about the nature of the WT mice used in each experiment. The methods described that the littermate floxed mice were used as controls in some experiments and "off-the-shelf" C57BL6/J mice were used in others and the term WT was used interchangeably to describe these different mice. The legends also did not detail what the sample size represented (individual mice or cultures) and what sex of mice were used for each experiment.

The wild type mice in this manuscript were all littermates with the exception of 3 individual mice. 2 straight “off the shelf” B6 wild type mice were used in the in vitro differentiation assay experiments. One non-littermate wild type B6 mouse from our colony was used for the blasting and proliferation experiments. For all three of these mice, the mice were age and sex-matched to the dLck-Cre HDAC3 cKO mice in the experiment to prevent any other variation. We have updated the Materials and methods and the figure legends to point out where non-littermate controls were used.

Overall, 106 male and 116 female mice were used in the experiments described in this manuscript. We didn’t put the number of male and female mice for each panel of each figure to prevent overcrowding of the figure legends. The sample size represents individual mice, this has been updated in the Materials and methods.

Comments for the authors:1. It is surprising that the filipin staining is lowered in T cells cultured in full serum media. I have conducted such studies in T cells cultured with HMG CoA reductase inhibitors and as unable to see decreases in filipin staining in T cells unless the T cells were cultured with HMG-CoA reductase inhibitors with serum-free media. This suggested to me that the media could be an important source of cholesterol for these cells. This aspect was not considered.

It is possible that the serum-supplemented RPMI we are using to culture the CD4^+^ T cells could be providing an important source of cholesterol to our cultured cells. Even with our current formulation, the Filipin signal is reduced even when normalized for cell volume, indicating that although the media may provide an important source of cholesterol, the HDAC3-deficient cells fail to maintain cholesterol levels to the level of their wild-type counterparts in culture. We hypothesize this is due to the overexpression of the cholesterol export proteins ABCA1 and ABCG1. Certainly, culture with serum-free media could exacerbate the difference in Filipin III signal when there is no longer any exogenous cholesterol to take up.

2. It would be good to gate the blasting and non-blasting cells with similar sized gates in the WT and KO T cells in the FSC/SSC plot before looking at filipin staining to rule out size effects on autofluorescence.

To rule out the effects of size on autofluorescence, we normalized the signals to the median FSC-W^3^ to approximate the volume of the cells. For the gating, lymphocytes (blasting and non-blasting) were both captured in the lymphocyte gate, which was applied universally to the WT and KO samples in each flow experiment to maintain consistency in the gating strategy. Collectively, these two measures allow us to differences in cholesterol content for Filipin III signal, and not due to changes in cell size or autofluorescence. For more information on the Filipin III analysis, please see the response to “essential revisions 3”

3. The finding that the deletion appears to be more efficient in the naïve mature T cells (25% still expressing), versus the memory T cells (75%), suggests that the HDAC3 defect may have given the remaining HDAC3+ T cells a selective advantage to enter the memory T cell pool as a result of homeostatic proliferation. This possibility was not considered.

We agree and have now added a sentence addressing that the HDAC3-sufficient cells appear to have a competitive advantage to populate the memory pool in this system (lines 139-141).

4. Legends are sometimes confusing. For example, for Figure 4 (are these HDAC- gated cells from dLck-Cre mice?). If so, what is the WT control. Do you have the internal WT control (HDAC3+ cells in the dLck-Cre floxed mice)? Why was this shown for some experiments and not others.

We have added a sentence in the figure 4 legend to clarify that these cells are HDAC3^-^ cells to eliminate the contaminating HDAC3^+^ cells. We omitted the HDAC3-sufficient cells from the graphs because most of the naïve HDAC3-sufficient CD4^+^ are recent thymic emigrants (RTEs), which have been shown to have significant functional impairment. We see results consistent with the previously described RTE phenotype, and thus they are not a proper WT control in these experiments. The WT controls are from completely separate WT mice.

References

Bensinger, S. J., Bradley, M. N., Joseph, S. B., Zelcer, N., Janssen, E. M., Hausner, M. A., … Tontonoz, P. (2008). LXR Signaling Couples Sterol Metabolism to Proliferation in the Acquired Immune Response. Cell, 134(1), 97–111. https://doi.org/10.1016/j.cell.2008.04.052

Chen, X., Zhao, Y., Guo, Z., Zhou, L., Okoro, E. U., and Yang, H. (2011). Transcriptional Regulation of ATP-binding Cassette Transporter A1 Expression by a Novel Signaling Pathway. The Journal of Biological Chemistry, 286(11), 8917. https://doi.org/10.1074/JBC.M110.214429

Cunningham, C. A., Helm, E. Y., and Fink, P. J. (2018). Reinterpreting recent thymic emigrant function: defective or adaptive? Current Opinion in Immunology, 51, 1. https://doi.org/10.1016/J.COI.2017.12.006

Stein, M., Dütting, S., Mougiakakos, D., Bösl, M., Fritsch, K., Reimer, D., … Mielenz, D. (2017). A defined metabolic state in pre B cells governs B-cell development and is counterbalanced by Swiprosin-2/EFhd1. Cell Death and Differentiation, 24(7), 1239. https://doi.org/10.1038/CDD.2017.52

Tan, H., Yang, K., Li, Y., Shaw, T. I., Wang, Y., Blanco, D. B., … Chi, H. (2017). Integrative Proteomics and Phosphoproteomics Profiling Reveals Dynamic Signaling Networks and Bioenergetics Pathways Underlying T Cell Activation. Immunity, 46(3), 488–503. https://doi.org/10.1016/j.immuni.2017.02.010

Tzur, A., Moore, J. K., Jorgensen, P., Shapiro, H. M., and Kirschner, M. W. (2011). Optimizing Optical Flow Cytometry for Cell Volume-Based Sorting and Analysis. PLOS ONE, 6(1), e16053. https://doi.org/10.1371/JOURNAL.PONE.0016053